# Understanding, Predicting and Better Resolving Q-Value Divergence in Offline-RL

**Yang Yue**[*1]    **Rui Lu**[*1]    **Bingyi Kang**[*2]    **Shiji Song**[1]    **Gao Huang**[†1]
[1] Department of Automation, BNRist, Tsinghua University    [2] ByteDance Inc.
{le-y22, r-lu21}@mails.tsinghua.edu.cn
bingykang@gmail.com {shijis, gaohuang}@tsinghua.edu.cn

## Abstract

The divergence of the Q-value estimation has been a prominent issue in offline reinforcement learning (offline RL), where the agent has no access to real dynamics. Traditional beliefs attribute this instability to querying out-of-distribution actions when bootstrapping value targets. Though this issue can be alleviated with policy constraints or conservative Q estimation, a theoretical understanding of the underlying mechanism causing the divergence has been absent. In this work, we aim to thoroughly comprehend this mechanism and attain an improved solution. We first identify a fundamental pattern, *self-excitation*, as the primary cause of Q-value estimation divergence in offline RL. Then, we propose a novel **S**elf-**E**xcite **E**igenvalue **M**easure (SEEM) metric based on Neural Tangent Kernel (NTK) to measure the evolving property of Q-network at training, which provides an intriguing explanation on the emergence of divergence. For the first time, our theory can reliably decide whether the training will diverge at an early stage, and even predict the order of the growth for the estimated Q-value, the model's norm, and the crashing step when an SGD optimizer is used. The experiments demonstrate perfect alignment with this theoretic analysis. Building on our insights, we propose to resolve divergence from a novel perspective, namely regularizing the neural network's generalization behavior. Through extensive empirical studies, we identify LayerNorm as a good solution to effectively avoid divergence without introducing detrimental bias, leading to superior performance. Experimental results prove that it can still work in some most challenging settings, i.e. using only 1% transitions of the dataset, where all previous methods fail. Moreover, it can be easily plugged into modern offline RL methods and achieve SOTA results on many challenging tasks. We also give unique insights into its effectiveness. Code can be found at https://offrl-seem.github.io.

## 1 Introduction

Off-policy Reinforcement Learning (RL) algorithms are particularly compelling for robot control [39; 13; 17] due to their high sample efficiency and strong performance. However, these methods often suffer divergence of value estimation, especially when value over-estimation becomes unmanageable. Though various solutions (*e.g.*, double q-network [19]) are proposed to alleviate this issue in the online RL setting, this issue becomes more pronounced in offline RL, where the agent can only learn from offline datasets with online interactions prohibited [32]. As a result, directly employing off-policy algorithms in offline settings confronts substantial issues related to value divergence [14; 30].

This raises a natural yet crucial question: *why is value estimation in offline RL prone to divergence, and how can we effectively address this issue?* Conventionally, *deadly triad* [46; 4; 47; 48] is

---

[*]Equal contribution. [†]Corresponding author.

37th Conference on Neural Information Processing Systems (NeurIPS 2023).

identified to be the main cause of value divergence. Specifically, it points out that a RL algorithm is susceptible to divergence at training when three components are combined, including off-policy learning, function approximation, and bootstrapping. As a special case of off-policy RL, offline RL further attributes the divergence to *distributional shift*: when training a Q-value function using the Bellman operator, the bootstrapping operation frequently queries the value of actions that are unseen in the dataset, resulting in cumulative extrapolation errors [14; 30; 33]. To mitigate this issue, existing algorithms either incorporate policy constraints between the learned policy and the behavior policy [42; 14; 53; 23; 30; 12; 51; 8], or make conservative/ensemble Q-value estimates [31; 38; 2; 41; 57; 15]. While these methods demonstrate some effectiveness by controlling the off-policy degree and bootstrapping—two components of the deadly triad—they often overlook the aspect of function approximation. This neglect leaves several questions unanswered and certain limitations unaddressed in current methodologies.

**How does divergence actually occur?** Most existing studies only provide analyses of linear functions as Q-values, which is usually confined to contrived toy examples [47; 4]. The understanding and explanation for the divergence of non-linear neural networks in practice are still lacking. Little is known about what transpires in the model when its estimation inflates to infinity and what essential mechanisms behind the deadly triad contribute to this phenomenon. We are going to answer this question from the perspective of function approximation in the context of offline RL.

**How to avoid detrimental bias?** Despite the effectiveness of policy constraint methods in offline RL, they pose potential problems by introducing detrimental bias. Firstly, they explicitly force the policy to be near the behavior policy, which can potentially hurt performance. Also, the trade-off between performance and constraint needs to be balanced manually for each task. Secondly, these methods become incapable of dealing with divergence when a more challenging scenario is encountered, *e.g.*, the transitions are very scarce. Instead, we are interested in solutions without detrimental bias.

Our study offers a novel perspective on the challenge of divergence in Offline-RL. We dissect the divergence phenomenon and identify *self-excitation*—a process triggered by the gradient update of the Q-value estimation model—as the primary cause. This occurs when the model's learning inadvertently elevates the target Q-value due to its inherent generalization ability, which in turn encourages an even higher prediction value. This mirage-like property initiates a self-excitation cycle, leading to divergence. Based on this observation, we develop theoretical tools with Neural Tangent Kernel (NTK) to provide in-depth understanding and precise prediction of Q-value divergence. This is the first work, to our knowledge, that accurately characterizes neural network dynamics in offline RL and explains the divergence phenomenon. More specifically, our contributions are three-fold:

- **Explanation:** We offer a detailed explanation of Q-value estimation divergence in offline RL settings with neural networks as non-linear estimators. We propose a novel metric, the **S**elf-**E**xcite **E**igenvalue **M**easure (SEEM), which is defined as the largest eigenvalue in the linear iteration during training, serving as an indicator of divergence. It also elucidates the fundamental mechanisms driving this divergence.
- **Prediction:** We can foresee the divergence through SEEM at an early stage of training before its estimation explodes. If divergence occurs, our theoretical framework is capable of predicting the growth pattern of the model's norm and the estimated Q-value. When the SGD optimizer is used, we can even correctly predict the specific timestep at which the model is likely to crash. Our experiments show our theoretical analysis aligns perfectly with reality.
- **Effective Solution:** Drawing from our findings, we suggest mitigating divergence by regularizing the model's generalization behavior. This approach allows us to avoid imposing strict constraints on the learned policy, while still ensuring convergence. Specifically, viewed through the lens of SEEM, we find that MLP networks with LayerNorm exhibit excellent local generalization properties while MLPs without LayerNorm don't. Then we conduct experiments to demonstrate that with LayerNorm in critic network, modern offline RL algorithms can consistently handle challenging settings when the offline dataset is significantly small or suboptimal.

## 2   Preliminaries

**Notation.** A Markov Decision Process (MDP) is defined by tuple $\mathcal{M} = (S, A, P, r, \nu)$, where $S \subseteq \mathbb{R}^{d_S}$ is the state space and $A \subseteq \mathbb{R}^{d_A}$ is the action space. $P : S \times A \mapsto \Delta(S)$ is the transition dynamics mapping from state-action pair to distribution in state space. $r : S \times A \mapsto \mathbb{R}$ is the reward function and $\nu \in \Delta(S)$ is the initial state distribution. An offline RL algorithm uses an offline dataset

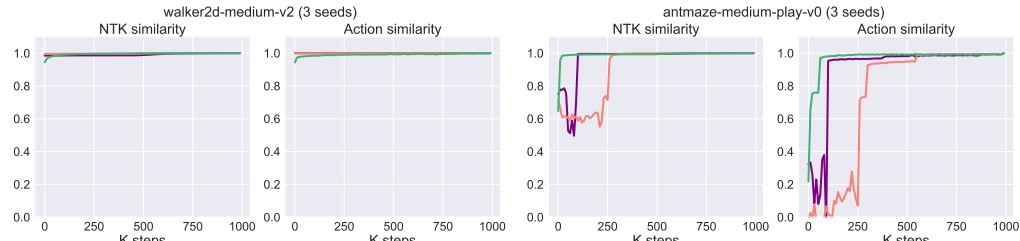

Figure 1: NTK similarity and action similarity along the training in two offline D4RL tasks. Different colors represent different seeds. The cosine similarity between the current step and the last step is computed. We can see all runnings reach a steady NTK direction and policy actions after a time $t_0$.

$D = \{(s_i, a_i, s'_i, r_i)\}_{i=1}^{M}$, which consists of $M$ finite samples from interactions with MDP $\mathcal{M}$. We write $s \in D$ if there exists some $k \in [M]$ such that $s_k = s$, similar for notation $(s, a) \in D$.

In this paper, we consider using a $L$-layer ReLU-activated MLP as the Q-value approximator. Each layer has parameter weight $\mathbf{W}_\ell \in \mathbb{R}^{d_{\ell+1} \times d_\ell}$ and bias $\mathbf{b}_\ell \in \mathbb{R}^{d_{\ell+1}}$. The activation function is ReLU: $\sigma(x) = \max(x, 0)$. Integer $d_\ell$ represents the dimensionality of the $\ell_{th}$ hidden layer, where $d_0 = d_S + d_A$ and $d_L = 1$ since a scalar estimation is required for Q-value. Denote $\boldsymbol{\theta} = \text{Vec}[\mathbf{W}_0, \mathbf{b}_0, \ldots, \mathbf{w}_L, \mathbf{b}_L]$ to be the vector of all learnable parameters. We use $\hat{Q}_{\boldsymbol{\theta}}(s, a) = f_{\boldsymbol{\theta}}(s, a)$ to denote the estimated Q-value using neural network $f_{\boldsymbol{\theta}}$ with parameter $\boldsymbol{\theta}$. Also, $\hat{\pi}_{\boldsymbol{\theta}}(s) = \arg\max_a \hat{Q}_{\boldsymbol{\theta}}(s, a)$ is its induced policy. Sometimes we will write $f_{\boldsymbol{\theta}}(\mathbf{X}) \in \mathbb{R}^M$ where $\mathbf{X} = [(s_1, a_1), \ldots, (s_M, a_M)]$ stands for the concatenation of all inputs $(s_i, a_i) \in D$ in dataset.

Considering a Q-learning-based offline RL algorithm that alternates between the following two steps:

$$\bar{Q} \leftarrow r(s, a) + \gamma \max_{a'} \hat{Q}_{\boldsymbol{\theta}}(s', a'), \quad \boldsymbol{\theta} \leftarrow \boldsymbol{\theta} - \eta \nabla_{\boldsymbol{\theta}} \left( \bar{Q} - \hat{Q}_{\boldsymbol{\theta}}(s, a) \right)^2, \tag{1}$$

where $\eta$ is the learning rate for updating the Q network. Practical algorithms might use its actor-critic form for implementation, we stick to this original form for theoretical analysis. In the following analysis, we call these two steps *Q-value iteration*.

**Neural Tangent Kernel (NTK).** NTK [21] is an important tool to analyze the dynamics and generalization of neural networks. Intuitively, it can be understood as a special class of kernel function, which measures the similarity between $\mathbf{x}$ and $\mathbf{x}'$, with dependency on parameter $\boldsymbol{\theta}$. Given a neural network $f_{\boldsymbol{\theta}}$ and two points $\mathbf{x}, \mathbf{x}'$ in the input space, the NTK is defined as $k_{\text{NTK}}(\mathbf{x}, \mathbf{x}') := \langle \nabla_{\boldsymbol{\theta}} f_{\boldsymbol{\theta}}(\mathbf{x}), \nabla_{\boldsymbol{\theta}} f_{\boldsymbol{\theta}}(\mathbf{x}') \rangle$, where the feature extraction is $\phi_{\boldsymbol{\theta}}(\mathbf{x}) := \nabla_{\boldsymbol{\theta}} f_{\boldsymbol{\theta}}(\mathbf{x})$. If we have two batches of input $\mathbf{X}, \mathbf{X}'$, we can compute the Gram matrix as $\mathbf{G}_{\boldsymbol{\theta}}(\mathbf{X}, \mathbf{X}') = \phi_{\boldsymbol{\theta}}(\mathbf{X})^\top \phi_{\boldsymbol{\theta}}(\mathbf{X}')$.

Intuitively speaking, NTK measures the similarity between two inputs through the lens of a non-linear network $f_{\boldsymbol{\theta}}$. Such similarity can be simply understood as the correlation between the network's value predictions of these two inputs. If $k_{\text{NTK}}(\mathbf{x}, \mathbf{x}')$ is a large value, it means these two points are "close" from the perspective of the current model's parameter. As a result, if the model's parameter changes near $\boldsymbol{\theta}$, say, from $\boldsymbol{\theta}$ to $\boldsymbol{\theta}' = \boldsymbol{\theta} + \Delta\boldsymbol{\theta}$, denote the prediction change for $\mathbf{x}$ as $\Delta f(\mathbf{x}) = f_{\boldsymbol{\theta}'}(\mathbf{x}) - f_{\boldsymbol{\theta}}(\mathbf{x})$ and for $\mathbf{x}'$ as $\Delta f(\mathbf{x}') = f_{\boldsymbol{\theta}'}(\mathbf{x}') - f_{\boldsymbol{\theta}}(\mathbf{x}')$. $\Delta f(\mathbf{x})$ and $\Delta f(\mathbf{x}')$ will be highly correlated, and the strength of the correlation is proportional to $k_{\text{NTK}}(\mathbf{x}, \mathbf{x}')$. This intuition can be explained from simple Taylor expansion, which gives $\Delta f(\mathbf{x}) = \Delta\boldsymbol{\theta}^\top \phi_{\boldsymbol{\theta}}(\mathbf{x})$ and $\Delta f(\mathbf{x}') = \Delta\boldsymbol{\theta}^\top \phi_{\boldsymbol{\theta}}(\mathbf{x}')$. These two quantities will be highly correlated if $\phi_{\boldsymbol{\theta}}(\mathbf{x})$ and $\phi_{\boldsymbol{\theta}}(\mathbf{x}')$ are well-aligned, which corresponds to large NTK value between $\mathbf{x}$ and $\mathbf{x}'$. To summary, $k_{\text{NTK}}(\mathbf{x}, \mathbf{x}')$ characterizes the subtle connection between $f_{\boldsymbol{\theta}}(\mathbf{x})$ and $f_{\boldsymbol{\theta}}(\mathbf{x}')$ caused by neural network's generalization.

## 3 Theoretical Analysis

In this section, we will investigate the divergence phenomenon of Q-value estimation and theoretically explains it for a more comprehensive understanding. Note that in this section our analysis and experiments are conducted in a setting that does not incorporate policy constraints and exponential moving average targets. Although the setting we analyze has discrepancies with real practice, our analysis still provides valueable insight, since the underlying mechanism for divergence is the same. We first focus on how temporal difference (TD) error changes after a single step of Q-value iteration in Equation (1). Our result is the following theorem.

**Theorem 1.** *Suppose that the network's parameter at iteration $t$ is $\boldsymbol{\theta}_t$. For each transition $(s_i, a_i, s_{i+1}, r_i)$ in dataset, denote $\boldsymbol{r} = [r_1, \ldots, r_M]^\top \in \mathbb{R}^M$, $\hat{\pi}_{\boldsymbol{\theta}_t}(s) = \arg\max_a \hat{Q}_{\boldsymbol{\theta}_t}(s, a)$. Denote $\boldsymbol{x}^*_{i,t} = (s_{i+1}, \hat{\pi}_{\boldsymbol{\theta}_t}(s_{i+1}))$. Concatenate all $\boldsymbol{x}^*_{i,t}$ to be $\boldsymbol{X}^*_t$. Denote $\boldsymbol{u}_t = f_{\boldsymbol{\theta}_t}(\boldsymbol{X}) - (\boldsymbol{r} + \gamma \cdot f_{\boldsymbol{\theta}_t}(\boldsymbol{X}^*_t))$ to be TD error vector at iteration $t$. The learning rate $\eta$ is infinitesimal. When the maximal point of $\hat{Q}_{\boldsymbol{\theta}_t}$ is stable as $t$ increases, we have the evolving equation for $\boldsymbol{u}_{t+1}$ as $\boldsymbol{u}_{t+1} = (\boldsymbol{I} + \eta \boldsymbol{A}_t)\boldsymbol{u}_t$, where $\boldsymbol{A}_t = (\gamma\phi_{\boldsymbol{\theta}_t}(\boldsymbol{X}^*_t) - \phi_{\boldsymbol{\theta}_t}(\boldsymbol{X}))^\top \phi_{\boldsymbol{\theta}_t}(\boldsymbol{X}) = \gamma\boldsymbol{G}_{\boldsymbol{\theta}_t}(\boldsymbol{X}^*_t, \boldsymbol{X}) - \boldsymbol{G}_{\boldsymbol{\theta}_t}(\boldsymbol{X}, \boldsymbol{X})$.*

$\boldsymbol{G}$ is defined in Section 2. Detailed proof is left in Appendix B. Theorem 1 states that when the policy action $\hat{\pi}_{\boldsymbol{\theta}_t}(s)$ that maximizes Q-values keep stable, leading to invariance of $\boldsymbol{X}^*_t$, each Q-value iteration essentially updates the TD error vector by a matrix $\boldsymbol{A}_t$, which is determined by a discount factor $\gamma$ and the NTK between $\boldsymbol{X}$ and $\boldsymbol{X}^*_t$.

### 3.1 Linear Iteration Dynamics and SEEM

Although Theorem 1 finds that TD update is a linear iteration under certain conditions, the dynamic is still complex in general. The policy $\hat{\pi}_{\boldsymbol{\theta}_t}(s)$ that maximizes Q-values may constantly fluctuate with $\boldsymbol{\theta}_t$ over the course of training, leading to variations of $\boldsymbol{X}^*_t$. Also, the kernel $\boldsymbol{G}_{\boldsymbol{\theta}}(\cdot, \cdot)$ has dependency on parameter $\boldsymbol{\theta}$. This causes non-linear dynamics of $\boldsymbol{u}_t$ and the model parameter vector $\boldsymbol{\theta}$. However, according to our empirical observation, we discover the following interesting phenomenon.

**Assumption 2. (Existence of Critical Point)** *There exists a time step $t_0$, a terminate kernel $\bar{k}(\cdot, \cdot)$ and stabilized policy state-action $\bar{\boldsymbol{X}}^*$ such that $\left\| \frac{k_t(\boldsymbol{x}, \boldsymbol{x}')}{\|\boldsymbol{x}\| \cdot \|\boldsymbol{x}'\|} - \frac{\bar{k}(\boldsymbol{x}, \boldsymbol{x}')}{\|\boldsymbol{x}\| \cdot \|\boldsymbol{x}'\|} \right\| = o(1)$ and $\left\| \boldsymbol{X}^*_{t+1} - \boldsymbol{X}^*_t \right\| = o(\eta)$ for any $\boldsymbol{x}, \boldsymbol{x}' \in \mathcal{S} \times \mathcal{A}$ when $t > t_0$. We also assume that $\left\| \boldsymbol{X}^*_t - \bar{\boldsymbol{X}}^* \right\| = o(1)$.*

This assumption states that the training dynamics will reach a steady state, where the policy $\hat{\pi}_{\boldsymbol{\theta}_t}$ stabilizes and NTK converges to a terminate direction. Usually, constant NTK requires the width of the network to be infinite[21]. But in our experiments[1], the convergence of NTK is still observed in all our environments despite finite width (see Figure 1). Results of more environments are attached in the appendix. It should be pointed out that NTK direction stability itself does not imply model convergence; rather, it only indicates that the model evolves in steady dynamics. Apart from NTK, we also observe that $\hat{\pi}_{\boldsymbol{\theta}_t}$ converges after a certain timestep $t_0$. Moreover, we find that $\hat{\pi}_{\boldsymbol{\theta}_t}$ always shifts to the border of the action set with max norm (see Figure 15). The reason for the existence of this critical point is elusive and still unknown. We hypothesize that it originates from the structure of gradient flow ensuring every trajectory is near an attractor in the parameter space and ends up in a steady state.

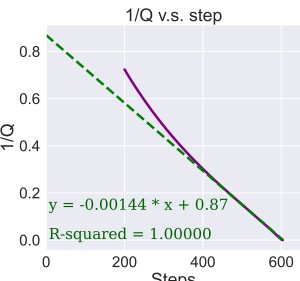

Figure 2: Prediction Ability - Linear decay of inverse Q-value with SGD and $L = 2$.

Based on the critical point assumption, we prove that the further evolving dynamic after the critical point is precisely characterized by linear iteration as the following theorem. We also explain why such a stable state can persist in appendix.

**Theorem 3.** *Suppose we use SGD optimizer for Q-value iteration with learning rate $\eta$ to be infinitesimal. Given iteration $t > t_0$, and $\boldsymbol{A} = \gamma\bar{\boldsymbol{G}}(\bar{\boldsymbol{X}}^*, \boldsymbol{X}) - \bar{\boldsymbol{G}}(\boldsymbol{X}, \boldsymbol{X})$, where $\bar{\boldsymbol{G}}$ is the Gram matrix under terminal kernel $\bar{k}$. The divergence of $\boldsymbol{u}_t$ is equivalent to whether there exists an eigenvalue $\lambda$ of $\boldsymbol{A}$ such that $\mathrm{Re}(\lambda) > 0$. If converge, we have $\boldsymbol{u}_t = (\boldsymbol{I} + \eta\boldsymbol{A})^{t-t_0} \cdot \boldsymbol{u}_{t_0}$. Otherwise, $\boldsymbol{u}_t$ becomes parallel to the eigenvector of the largest eigenvalue $\lambda$ of $\boldsymbol{A}$, and its norm diverges to infinity at following order*

$$\|\boldsymbol{u}_t\|_2 = O\left( \frac{1}{(1 - C'\lambda\eta t)^{L/(2L-2)}} \right) \tag{2}$$

*for some constant $C'$ to be determined and $L$ is the number of layers of MLP. Specially, when $L = 2$, it reduces to $O\left( \frac{1}{1 - C'\lambda\eta t} \right)$.*

Theorem 3 predicts the divergence by whether $\lambda_{\max}(\boldsymbol{A})$ is greater to 0. We term this *divergence detector* $\lambda_{\max}(\boldsymbol{A})$ as **S**elf-**E**xcite **E**igenvalue **M**easure, or SEEM. Essentially, Theorem 3 states that

---

[1]To affirm the practicability of our analysis, we conduct experiments on a widely used offline RL benchmark D4RL [10], and utilize a 3-Layer MLP instead of a toy example. More details can be found in the appendix.

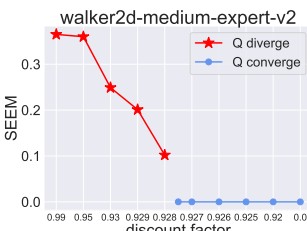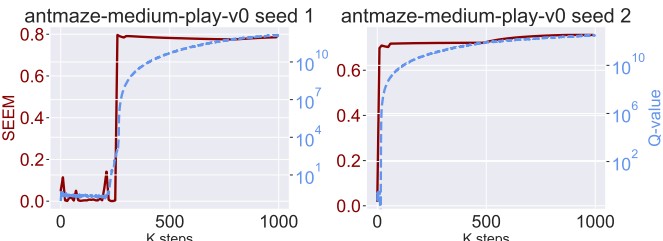

Figure 3: The divergence indication property of SEEM. The left figure shows the SEEM value with respect to different discount factors $\gamma$. Theorem 3 states that larger $\gamma$ has a larger SEEM value. The star point means the model's prediction eventually exceeds a large threshold ($10^6$) in training, which means diverging. We can see that positive SEEM is perfectly indicative of the divergence. This is also true within the training process. From the middle and right figures, we can see that the prediction Q-value (in blue) is stable until the normalized kernel matrix's SEEM (in red) rises up to a large positive value, then we can observe the divergence of the model.

we can monitor SEEM value to know whether the training will diverge. To validate this, we conduct experiments with different discount factor $\gamma$. The first term in $\boldsymbol{A}$ is linear to $\gamma$, thus larger $\gamma$ is more susceptible to diverging. This is exactly what we find in reality. In Figure 3, we can see that experiments with large $\gamma$ have positive SEEM, and the training eventually diverges ($\bigstar$). Moreover, SEEM can also faithfully detect the trend of divergence during the training course. On the right of Figure 3, we can see that the surge of SEEM value is always in sync with the inflation of estimation value. All these results corroborate our theoretical findings.

## 3.2 Prediction Ability

In Section 3.1, we have demonstrated that SEEM can reliably predict whether the training will diverge. In fact, our theory is able to explain and predict many more phenomena, which are all exactly observed empirically. Here we list some of them, and the detailed proofs are left in supplementary.

➢ **Terminating (Collapsing) Timestep.** Theorem 3 predicts that with an SGD optimizer and 2-layer MLP, the inverse of Q-value decreases linearly along the timestep, implying a terminating timestep $T = \frac{1}{C'\lambda\eta}$. The Q-value estimation will approach infinite very quickly beside $T$, and ends in singularity because the denominator becomes zero. From Figure 2, we can see that it is true. The terminal Q-value prediction's inverse *does* decay linearly, and we can predict when it becomes zero to hit the singularity. Specifically, we use the data from 450th to 500th step to fit a linear regression, obtaining the green dotted line $1/\|\boldsymbol{u}_t\| \approx -1.44 \times 10^{-3}t + 0.87$, which predicts a singular point at $605_{th}$ iteration. We then continue training and find that the model indeed collapses at the very point, whose predicted value and parameter become NaN.

➢ **Linear Norm Growth for Adam.** While Theorem 3 studies SGD as the optimizer, Adam is more adopted in real practice. Therefore, we also deduce a similar result for the Adam optimizer.

**Theorem 4.** *Suppose we use Adam optimizer for Q-value iteration and all other settings are the same as Theorem 3. After $t > t_0$, the model will diverge if and only if $\lambda_{\max}(\boldsymbol{A}) > 0$. If it diverges, we have $\|\boldsymbol{\theta}_t\| = \eta\sqrt{P}t + o(t)$ and $\|\boldsymbol{u}_t\| = \Theta(t^L)$ where $P$ and $L$ are the number of parameters and the number of layers for network $f_{\boldsymbol{\theta}}$, respectively.*

Again, the detailed proof is left in the supplementary. Theorem 4 indicates that with a Adam optimizer, the norm of the network parameters grows linearly and the predicted Q-value grows as a polynomial of degree $L$ along the time after a critical point $t_0$. We verify this theorem in D4RL environments in Figure 4. We can see that the growth of the norm $\|\boldsymbol{\theta}_t\|$ exhibits a straight line after a critic point $t_0$. Moreover, we can see $\log Q \approx 3\log t + c$, which means Q-value prediction grows cubically with time. Number 3 appears here because we use 3-layer MLP for value approximation. All these findings corroborate our theory, demonstrating our ability to accurately predict the dynamic of divergence in offline RL.

## 3.3 Discussions

**Interpretation of SEEM and explain the self-excitation** The intuitive interpretation for $\lambda_{\max}(\boldsymbol{A})$ is as below: Think about only one sample $\boldsymbol{x} = (s_0, a_0)$ and its stabilized next step state-action

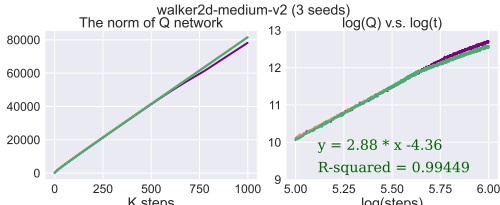
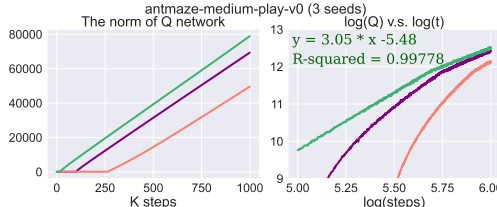

Figure 4: Prediction ability - Linear growth of network parameter's norm and polynomial growth of predicted Q-value with an Adam optimizer. Please note that in the 2th and 4th figures, both Q-value and steps have been taken the logarithm base 10. Different color represents different seeds.

$\boldsymbol{x}^* = (s_1, a_1)$. Without loss of generality, we assume $f_{\boldsymbol{\theta}_t}(\boldsymbol{x}) < r + \gamma f_{\boldsymbol{\theta}_{t+1}}(\boldsymbol{x}^*)$. If SEEM is positive, we have $\gamma \boldsymbol{G}_{\boldsymbol{\theta}_t}(\boldsymbol{x}^*, \boldsymbol{x})$ is larger than $\boldsymbol{G}_{\boldsymbol{\theta}_t}(\boldsymbol{x}, \boldsymbol{x})$. Recall that $\boldsymbol{G}_{\boldsymbol{\theta}_t}(\boldsymbol{x}^*, \boldsymbol{x})$ depicts the strength of the bond between $\boldsymbol{x}$ and $\boldsymbol{x}^*$ because of generalization. So we know that, whren updating the value of $f_{\boldsymbol{\theta}}(\boldsymbol{x})$ towards $r + \gamma f_{\boldsymbol{\theta}_{t+1}}(\boldsymbol{x}^*)$, the Q-value iteration inadvertently makes $f_{\boldsymbol{\theta}_{t+1}}(\boldsymbol{x}^*)$ increase even more than the increment of $f_{\boldsymbol{\theta}_{t+1}}(\boldsymbol{x})$. Consequently, the TD error $r + \gamma f_{\boldsymbol{\theta}}(\boldsymbol{x}^*) - f_{\boldsymbol{\theta}}(\boldsymbol{x})$ expands instead of reducing, due to the target value moving away faster than predicted value, which encourages the above procedure to repeat. This forms a positive feedback loop and causes self-excitation. Such mirage-like property causes the model's parameter and its prediction value to diverge.

**Stability of policy action** $\hat{\pi}_{\boldsymbol{\theta}_t}(s)$**.** Previous Theorem 1 and Theorem 3 rely on the stability of the policy action $\hat{\pi}_{\boldsymbol{\theta}_t}(s)$ after $t_0$, and here we further elucidate why such stability occurs. Critic networks without normalization have a tendency to output large values for extreme points, *i.e.*, action $A_{ex}$ at the boundary of action space (see detailed explanation in Section 4), making these extreme points easily become policy actions during optimization. We have observed that policy actions tend to gravitate toward extreme points when Q-value divergence takes place (see Figure 15), accounting for the stability of policy action $\hat{\pi}_{\boldsymbol{\theta}_t}(s)$. Consequently, as aforementioned, a loop is formed: $\hat{Q}(s, a)$ keeps chasing $\hat{Q}(s', A_{ex})$, leading the model's parameter to diverge along certain direction to infinity.

**Relations to Linear Setting.** Since the linear regression model can be regarded as a special case for neural networks, our analysis is also directly applicable to linear settings by plugging in $\boldsymbol{G}(\boldsymbol{X}, \boldsymbol{X}) = \boldsymbol{X}^\top \boldsymbol{X}$, which reduce to study of deadly triad in linear settings. Therefore, our analysis fully contains linear case study and extends it to non-linear value approximation. In Appendix F.5, we present that our solution (Section 4) can solve the divergence in Baird's Counterexample, which was first to show the divergence of Q-learning with linear approximation.

**Similarities and Differences to Previous works.** Our work shares similar observations of the connection between Q-value divergence and feature rank collapse with DR3 [29]. However, our work is different in the following aspects. First, DR3 attributes feature rank collapse to implicit regularization where $L(\theta) = 0$, and it require the label noise $\varepsilon$ from SGD. Actually, such near-zero critic loss assumption is mostly not true in the real practice of RL. Conversely, SEEM provides a mechanism behind feature rank collapse and Q-value divergence from the perspective of normalization-free network's pathological extrapolation behavior. It is applicable to more general settings and provides new information about this problem. Moreover, we formally prove in Theorems 3 and 4 that the model's parameter $\theta$ evolves linearly for the Adam optimizer, and collapses at a certain iter for SGD by solving an ODE. This accurate prediction demonstrates our precise understanding of neural network dynamics and is absent in previous work like DR3. Last, the solution to value divergence by DR3 involves searching hyperparameter tuning for $c_0$ and also introduces extra computation when computing the gradient of $\phi(s', a')$ to get $\overline{\mathcal{R}}_{exp}(\theta)$. Our method (elaborated in Section 4) is free of these shortcomings by simple LayerNorm.

## 4 Reducing SEEM By Normalization

In Section 3, we have identified SEEM as a measure of divergence, with self-excitation being the primary catalyst for such divergence. In essence, a large SEEM value arises from the improper link between the dataset inputs and out-of-distribution data points. To gain an intuitive grasp, we visualize the NTK value in a simple 2-dimensional input space for a two-layer MLP in Figure 5. We designate a reference point $\boldsymbol{x}_0 = (0.1, 0.2)$ and calculate the NTK value $\boldsymbol{G}_{\boldsymbol{\theta}}(\boldsymbol{x}_0, \boldsymbol{x})$ for a range of $\boldsymbol{x}$ in the input domain. The heatmap displays high values at the boundaries of the input range,

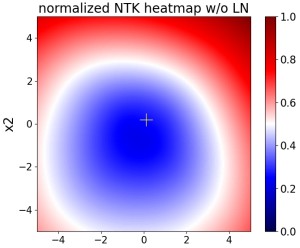
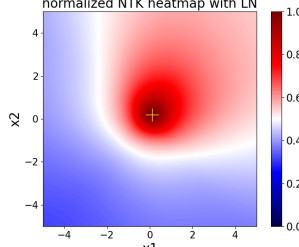

Figure 5: The ***normalized*** NTK map for 2-layer ReLU MLP with and without layernorm. The input and hidden dimensions are 2 and 10,000. We can see that for the MLP without LayerNorm, prediction change at $x_0$ has a dramatic influence on points far away like $x = (4, 4)$, meaning a slight change of $f(x_0)$ will change $f(x)$ dramatically ($\sim 4.5\times$). However, MLP equipped with layernorm exhibits good local property.

which suggests that even a slight increase (or decrease) in the prediction value at $x_0$ will result in an amplified increase (*e.g.*, $4.5\times$) in the prediction value at a distant point(*e.g.*, $(4, 4)$). Note that indeed the absolute NTK surrounding $x_0$ is positive. However, values farther from $x_0$ are significantly larger. As a result, the NTK around $x_0$ is minimal and normalized to a value close to zero. Thus, the value predictions of the dataset sample and extreme points have large NTK value and exhibit a strange but strong correlation. As Q-value network is iteratively updated, it also tend to output large values for these extreme points, which makes extreme points easily become policy actions in optimization.

This abnormal behavior contradicts the typical understanding of a "kernel function" which usually diminishes with increasing distance, which represents the MLP's inherent limitation in accurate extrapolation. This indicates an intriguing yet relatively under-explored approach to avoid divergence: *regularizing the model's generalization on out-of-distribution predictions*. The main reason for such an improperly large kernel value is that the neural network becomes a linear function when the input's norm is too large [54]. A more detailed explanation about linearity can be found at Appendix G. Therefore, a simple method to accomplish this would be to insert a LayerNorm prior to each non-linear activation. We conduct a similar visualization by equipping the MLP with LayerNorm [3]. As shown in Figure 5 right, the value reaches its peak at $x_0$ and diminishes as the distance grows, demonstrating excellent local properties for well-structured kernels.

Such architectural change is beneficial for controlling the eigenvalue of $A$ by reducing the value of improperly large entries since matrix inequality tells us $\lambda_{\max}(A) \leq \|A\|_F$. Therefore, it is expected that this method can yield a small SEEM value, ensuring the convergence of training with minimal bias on the learned policy. We also provide theoretical justification explaining why LayerNorm results in a lower SEEM value in the supplementary material. This explains why LayerNorm, as an empirical practice, can boost performance in previous online and offline RL studies [41; 5; 25; 28]. Our contribution is thus two-fold: 1) We make the first theoretical explanation for how LayerNorm mitigates divergence through the NTK analysis above. 2) We conduct thorough experiments to empirically validate its effectiveness, as detailed in Section Section 5.

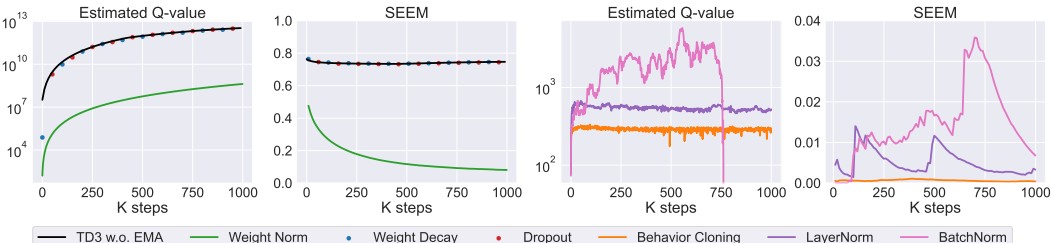

Figure 6: The effect of various regularizations and normalizations on SEEM and Q-value. LayerNorm yields a low SEEM and achieves stable Q-learning. Given the substantial disparity in the y-axis range among various regularizations, we present the results using two separate figures.

To validate the effectiveness of LayerNorm in practical offline RL settings, we select the walker2d-medium-expert-v2 task in D4RL to showcase how Q-value and SEEM evolves as the training proceeds. For comparative analysis, we also consider four popular regularization and normalization techniques, including BatchNorm [20], WeightNorm [44], dropout [45], and weight decay [35]. We

Table 1: Averaged normalized scores of the last ten consecutive checkpoints on Antmaze tasks over 10 seeds. diff-QL [51] reports the best score during the whole training process in its original paper. We rerun its official code for the average score. We also compare our method with diff-QL by the best score metrics in the parenthesis. The best score of diff-QL is directly quote from its paper.

| Dataset | TD3+BC | IQL | MSG | sfBC | diff-QL | ours |
|---|---|---|---|---|---|---|
| antmaze-umaze-v0 | 40.2 | 87.5 | **98.6** | 93.3 | 95.6 (96.0) | $94.3 \pm 0.5$ (97.0) |
| antmaze-umaze-diverse-v0 | 58.0 | 62.2 | 76.7 | 86.7 | 69.5 (84.0) | $\mathbf{88.5 \pm 6.1}$ (95.0) |
| antmaze-medium-play-v0 | 0.2 | 71.2 | 83.0 | **88.3** | 0.0 (79.8) | $85.6 \pm 1.7$ (92.0) |
| antmaze-medium-diverse-v0 | 0.0 | 70.0 | 83.0 | **90.0** | 6.4 (82.0) | $83.9 \pm 1.6$ (90.7) |
| antmaze-large-play-v0 | 0.0 | 39.6 | 46.8 | 63.3 | 1.6 (49.0) | $\mathbf{65.4 \pm 8.6}$ (74.0) |
| antmaze-large-diverse-v0 | 0.0 | 47.5 | 58.2 | 41.7 | 4.4 (61.7) | $\mathbf{67.1 \pm 1.8}$ (75.7) |
| average | 16.4 | 63.0 | 74.4 | 77.2 | 29.6 (75.4) | **80.8** (87.4) |

use TD3 as the baseline algorithms. To simplify our analysis, the target network is chosen to be the current Q-network with gradient stop instead of an Exponential Moving Average (EMA) of past networks. In our experiments, it is observed that LayerNorm and WeightNorm constrain SEEM and restrain divergence. However, weight decay and dropout did not yield similar results. As illustrated in Figure 6, weight decay and dropout evolve similar to the unregularized TD3 w.o. EMA baseline. WeightNorm reduces SEEM by a clear margin compared to the baseline, thus demonstrating slighter divergence. In the right figure, we observe that behavior cloning effectively lowers SEEM at the cost of introducing significant explicit bias. Importantly, LayerNorm achieves both a low SEEM and stable Q-values without necessitating explicit policy constraints. A notable outlier is BatchNorm. BatchNorm attains a relatively low maximum eigenvalue at the beginning of training but experiences an increase over time. Correspondingly, the Q-curve displays substantial oscillations. This instability could be ascribed to the shared batch normalization statistics between the value network and the target network, despite their input actions originating from distinct distributions [6].

## 5 Agent Performance

Previous state-of-the-art offline RL algorithms have performed exceptionally well on D4RL Mujoco Locomotion tasks, achieving an average score above 90 [2; 41]. In this section, we compare our method with various baselines on two difficult settings, *i.e.*, Antmaze and X% Mujoco task, to validate the effectiveness of LayerNorm in offline settings. Further experiments can be found at Appendix F.

### 5.1 Standard Antmaze Dataset

In Antmaze tasks characterized by sparse rewards and numerous suboptimal trajectories, the prior successful algorithms either relies on in-sample planning, such as weighted regression [26; 8], or requires careful adjustment the number of ensembles per game [15]. Algorithms based on TD3 or SAC failed to achieve meaningful scores by simply incorporating a behavior cloning (BC) term [12]. Even Diff-QL, which replaces TD3+BC's Gaussian policy with expressive diffusion policies to capture multi-modal behavior [51], continues to struggle with instability, leading to inferior performance.

We first show policy constraint (BC) is unable to control q-value divergence while performing well in some challenging environments, by using Antmaze-large-play as an running example. We conduct experiments with Diff-QL by varying the BC (*i.e.*, diffusion loss) coefficient from 0.5 to 10. As shown in Figure 7, when the policy constraint is weak (BC 0.5), it initially achieves a decent score, but as the degree of off-policy increases, the value starts to diverge, and performance drops to zero. Conversely, when the policy constraint is too strong (BC 10), the learned policy cannot navigate out of the maze due to suboptimal data, and performance remains zero. In contrast, simply incorporating LayerNorm into Diff-QL, our method ensures stable value convergence under less restrictive policy constraints (BC 0.5). This results in consistently stable performance in the challenging Antmaze-large-play task. We report the overall results in Table 1, which shows that our method consistently maintains stable average scores across six distinct environments. Additionally, we are able to boost the highest score, while adopting the same evaluation metric used by Diff-QL. Ultimately, our method achieves an average score of 80.8 on Antmaze tasks, exceeding the performance of previous SOTA methods. In summary, our experimental findings demonstrates within suboptimal datasets, an overly strong policy constraint is detrimental, while a weaker one may lead to value divergence. LayerNorm proves effective in maintaining stable value convergence under less restrictive policy constraints,

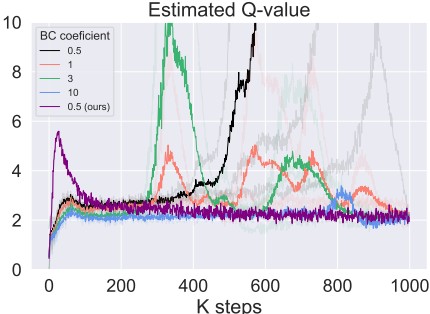
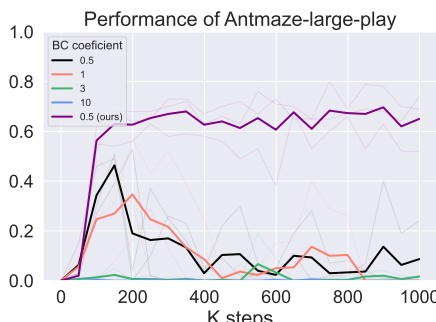

Figure 7: The detrimental bias in policy constraint. Although strong policy constraints succeed in depressing divergence, constraining the policy around sub-optimal behavior sacrifices performance. With LayerNorm to regularize SEEM without introducing a too strong bias, our method can achieve both stable Q-value convergence and excellent performance. The shallow curves represent the trajectory of individual seeds, while the darker curve denotes the aggregate across 3 seeds.

resulting in an excellent performance. The ablation study regarding the specific configuration of adding LayerNorm can be found at Appendix F.4. The effect of other regularizations in Antmaze can be found at F.3. We also experimented with the regularizer proposed in DR3 [29]. Despite conducting a thorough hyperparameter search for the coefficient, it was observed that it had no substantive impact on enhancing the stability and performance of diff-QL.

## 5.2  $X\%$ **Offline Dataset**

Past popular offline RL algorithms primarily require transitions rather than full trajectories for training [40; 31; 26; 12; 7; 51]. A superior algorithm should extrapolate the value of the next state and seamlessly stitch together transitions to form a coherent trajectory. However, commonly used benchmarks such as D4RL and RL Unplugged [10; 16] contain full trajectories. Even with random batch sampling, complete trajectories help suppress value divergence since the value of the next state $s_{t+1}$ will be directly updated when sampling $\{s_{t+1}, a_{t+1}, s_{t+2}, r_{t+1}\}$. On the one hand, working with offline datasets consisting solely of transitions enables a more authentic evaluation of an algorithm's stability and generalization capabilities. On the other hand, it is particularly relevant in real-world applications such as healthcare and recommendation systems where it is often impossible to obtain complete patient histories.

We first evaluate the performance of offline algorithms within a transition-based scenario. We construct transition-based datasets by randomly sampling varying proportions ($X\%$) from the D4RL Mujoco Locomotion datasets. Here, we set several levels for $X \in \{1, 10, 50, 100\}$. When X equals 100, it is equivalent to training on the original D4RL dataset. As X decreases, the risk of encountering out-of-distribution states and actions progressively escalates. To ensure a fair comparison, we maintain the same subset across all algorithms. We evaluate every algorithm on Walker2d, hopper, and Halfcheetah tasks with medium, medium replay, and medium expert levels. For comparison, we report the total score on nine Mujoco locomotion tasks and the standard deviation over 10 seeds. Figure 8 reveal a marked drop in performance for all popular offline RL algorithms when the dataset is reduced to 10%. When the transition is very scarce (1%), all baselines achieve

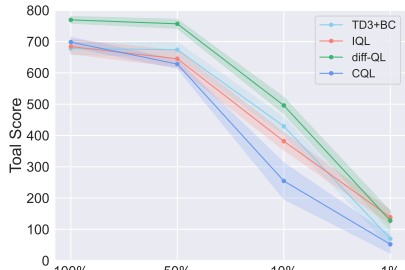

Figure 8: The performance of offline RL algorithms with the varying $X\%$ Mujoco Locomotion dataset. As the number of transitions decreases, all algorithms have a dramatic drop in performance.

about only 50 to 150 total points on nine tasks. Further, we observed value divergence in all algorithms, even for algorithms based on policy constraints or conservative value estimation.

Subsequently, we demonstrate the effectiveness of LayerNorm in improving the poor performance in X% datasets. By adding LayerNorm to the critic, value iteration for these algorithms becomes non-expansive, ultimately leading to stable convergence. As Figure 9 depicts, under 10% and 1% dataset, all baselines are greatly improved by LayerNorm. For instance, under the 1% dataset, LayerNorm

enhances the performance of CQL from 50 points to 300 points, marking a 6x improvement. Similarly, TD3+BC improves from 70 to 228, representing approximately a 3x improvement. This empirical evidence underscores the necessity of applying normalization to the function approximator for stabilizing value evaluation, particularly in sparse transition-based scenarios. Adding LayerNorm is shown to be more effective than relying solely on policy constraints. Another notable observation from Figure 9 is that with sufficient data (the 100% dataset) where previous SOTA algorithms have performed excellently, LayerNorm only marginally alters the performance of algorithms. This suggests that LayerNorm could serve as a universally applicable plug-in method, effective regardless of whether the data is scarce or plentiful.

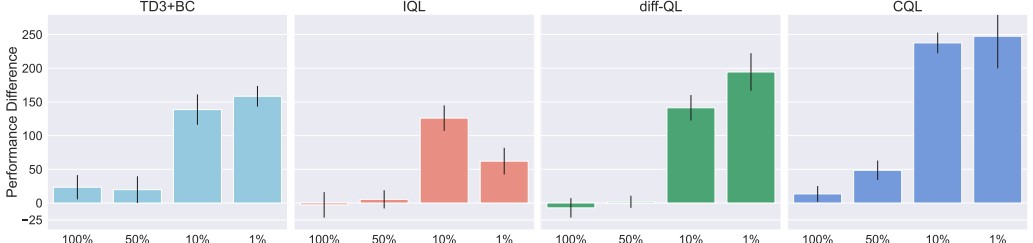

Figure 9: The performance difference between baseline with LayerNorm and without it using the same $X\%$ dataset. The error bar represents the standard deviation over 10 seeds.

### 5.3 Online RL Experiments - LayerNorm Allows Online Methods without EMA.

It is natural to ask whether our analysis and solution are also applicable to online settings. Previous works [34; 13; 17; 61] has empirically or theoretically shown the effect of EMA in stabilizing Q-value and prevent divergence in online setting. To answer the question, We tested whether LayerNorm can replace EMA to prevent value divergence. We conducted experiments in two environments in online settings: Hopper and Walker. Please refer to Figure 11 for the curves of return. Surprisingly, we discovered that LayerNorm solution allows the SAC without EMA to perform equivalently well as the SAC with EMA. In contrast, SAC without EMA and LayerNorm behaved aimlessly, maintaining near-zero scores. It reveals LayerNorm (or further regularizations discovered by SEEM later) have the potential of allowing online DQN-style methods to step away from EMA/frozen target approaches altogether towards using the same online and target networks. These results partially reveals potential implications of SEEM in an online learning setting.

## 6 Conclusion and Limitations

In this paper, we delve into the Q-value divergence phenomenon of offline RL and gave a comprehensive theoretical analysis and explanation for it. We identified a fundamental process called *self-excitation* as the trigger for divergence, and propose an eigenvalue measure called SEEM to reliably detect and predict the divergence. Based on SEEM, we proposed an orthogonal perspective other than policy constraint to avoid divergence, by using LayerNorm to regularize the generalization of the MLP neural network. We demonstrated empirical and theoretical evidence that our method introduces less bias in learned policy, which has better performance in various settings. Moreover, the SEEM metric can serve as an indicator, guiding future works toward further engineering that may yield adjustments with even better performance than the simple LayerNorm. Despite the promising results, our study has some limitations. we have not incorporated the impact of Exponential Moving Average (EMA) into our analysis. Also, the existence of critical point is still not well-understood. Moreover, our analysis does not extend to online RL, as the introduction of new transitions disrupts the stability of the NTK direction. All these aspects are left for future work.

## 7 Acknowledgement

This work is supported by the National Key R&D Program of China (2022ZD0114900).

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

# A  Related Works

**Value Divergence with Neural Network.** In online reinforcement learning (RL), off-policy algorithms that employ value function approximation and bootstrapping can experience value divergence, a phenomenon known as the deadly triad [46; 4; 47; 48; 11]. Deep Q-Networks (DQN) typify this issue. As they employ neural networks for function approximation, they are particularly susceptible to Q-value divergence [19; 13; 48; 60]. Past research has sought to empirically address this divergence problem through various methods, such as the use of separate target networks [39] and Double-Q learning [19; 13]. Achiam *et al.* [1] analyze a linear approximation of Q-network to characterizes the diverge, while CrossNorm [6] uses a elaborated version of BatchNorm [20] to achieve stable learning. Value divergence becomes even more salient in offline RL, where the algorithm learns purely from a fixed dataset without additional environment interaction [14; 30]. Much of the focus in the field of offline RL has been on controlling the extent of off-policy learning, *i.e.*, policy constraint [12; 40; 26; 14; 53; 51; 8; 42]. Several previous studies [41; 5; 25; 28] have empirically utilized LayerNorm to enhance performance in online and offline RL. These empirical results partially align with the experimental section of our work. However, our study makes a theoretical explanation for how LayerNorm mitigates divergence through the NTK analysis. Specifically, we empirically and theoretically illustrate how LayerNorm reduces SEEM. In addition to LayerNorm, our contribution extends to explaining divergence and proposing promising solutions from the perspective of reducing SEEM. Specially, we discover that WeightNorm can also be an effective tool and explain why other regularization techniques fall short. Finally, we perform comprehensive experiments to empirically verify the effectiveness of LayerNorm on the %X dataset, a practical setting not explored in previous work. Thus, our contributions are multifaceted and extend beyond the mere application of LayerNorm.

**Offline RL.** Traditional RL and bandit methods have a severe sample efficiency problem [32; 14; 36; 37]; offline RL aims to tackle this issue. However, offline RL presents significant challenges due to severe off-policy issues and extrapolation errors. Some existing methods focuses on designs explicit or implicit policy regularizations to minimize the discrepancy between the learned and behavior policies. For example, TD3+BC [13; 12; 55; 56] directly adds a behavior cloning loss to mimic the behavior policy, Diffusion-QL [51] further replace the BC loss with a diffusion loss and using diffusion models as the policy. AWR-style [42; 52; 7] and IQL [26] impose an implicit policy regularization by maximizing the weighted in-distribution action likelihood. Meanwhile, some other works try to alleviate the extrapolation errors by modifying the policy evaluation procedure. Specifically, CQL [31] penalizes out-of-distribution actions for having higher Q-values, while MISA [38] regularizes both policy improvement and evaluation with the mutual information between state-action pairs of the dataset. Alternatively, decision transformer (DT) [9] and trajectory transformer [22] cast offline RL as a sequence generation problem, which are beyond the scope of this paper. Despite the effectiveness of the above methods, they usually neglect the effect of function approximator and are thus sensitive to hyperparameters for trading off performance and training stability. Exploration into the function approximation aspect of the deadly triad is lacking in offline RL. Moreover, a theoretical analysis of divergence in offline RL that does not consider the function approximator would be inherently incomplete. We instead focus on this orthogonal perspective and provide both theoretical understanding and empirical solution to the offline RL problem.

# B  Proof of Main Theorems

Before proving our main theorem, we first state an important lemma.

**Lemma 1.** *For L-layer ReLU-activate MLP and fixed input $\boldsymbol{x}, \boldsymbol{x}'$, assume the biases are relatively negligible compared to the values in the hidden layers. If we scale up every parameter of $f_{\boldsymbol{\theta}}$ to $\lambda$ times, namely $\boldsymbol{\theta}' = \lambda\boldsymbol{\theta}$, then we have following equations hold for any $\boldsymbol{x}, \boldsymbol{x}' \in \mathcal{S} \times \mathcal{A}$*

$$\left\| f_{\boldsymbol{\theta}'}(\boldsymbol{x}) - \lambda^L f_{\boldsymbol{\theta}}(\boldsymbol{x}) \right\| = o(1) \cdot \left\| f_{\boldsymbol{\theta}'}(\boldsymbol{x}) \right\|$$

$$\left\| \nabla_{\boldsymbol{\theta}} f_{\boldsymbol{\theta}'}(\boldsymbol{x}) - \lambda^{L-1} \nabla_{\boldsymbol{\theta}} f_{\boldsymbol{\theta}}(\boldsymbol{x}) \right\| = o(1) \cdot \left\| \nabla_{\boldsymbol{\theta}} f_{\boldsymbol{\theta}'}(\boldsymbol{x}) \right\|,$$

$$\left| k_{\boldsymbol{\theta}'}(\boldsymbol{x}, \boldsymbol{x}') - \lambda^{2(L-1)} k_{\boldsymbol{\theta}}(\boldsymbol{x}, \boldsymbol{x}') \right| = o(1) \cdot k_{\boldsymbol{\theta}'}(\boldsymbol{x}, \boldsymbol{x}').$$

*Proof.* Recursively define

$$\boldsymbol{z}_{\ell+1} = \boldsymbol{W}_\ell \tilde{\boldsymbol{z}}_\ell + \boldsymbol{b}_\ell, \quad \tilde{\boldsymbol{z}}_0 = \boldsymbol{x}$$

$$\tilde{z}_\ell = \sigma(z_\ell).$$

Then it is easy to see that if we multiply each $W_\ell$ and $b_\ell$ by $\lambda$, denote the new corresponding value to be $z'_\ell$ and note that the bias term is negligible, we have

$$
\begin{aligned}
z'_1 =&\lambda z_1 \\
z'_2 =&\lambda W_2 \sigma(z'_1) + \lambda b_2 \\
=&\lambda^2 (W_2 \tilde{z}_1 + b_2) + (\lambda - \lambda^2) b_2 \\
=&\lambda^2 z_2 + (\lambda - \lambda^2) b_2 \\
=&\lambda^2 z_2 + \epsilon_2 \qquad\qquad\qquad \text{(negilible bias } \|\epsilon_2\| = o(\lambda^2)) \\
&\cdots \\
z'_{\ell+1} =&\lambda W_\ell \sigma(z'_\ell) + \lambda b_{\ell+1} \\
=&\lambda W_\ell \sigma(\lambda^\ell z_\ell + \epsilon_\ell) + \lambda b_{\ell+1} \\
=&\lambda^{\ell+1} W_\ell \sigma(z'_{\ell+1}) + \lambda b_{\ell+1} + \lambda W_\ell \delta_\ell \qquad (\delta_\ell = \sigma(\lambda^\ell z_\ell + \epsilon_\ell) - \sigma(z'_{\ell+1})) \\
=&\lambda^{\ell+1} z_{\ell+1} + (\lambda - \lambda^{\ell+1}) b_{\ell+1} + \lambda W_\ell \delta_\ell
\end{aligned}
$$

Note that $\lambda \|w_\ell \delta_\ell\| = \lambda \|W_\ell (\sigma(\lambda^\ell z_\ell + \epsilon_\ell) - \sigma(z'_{\ell+1}))\| \leq \lambda \|W_\ell\| \cdot \|(\lambda^\ell z_\ell + \epsilon_\ell) - z'_{\ell+1}\|$, the last step is because $\sigma(\cdot)$ is 1-Lipshitz, and we have $(\lambda^\ell z_\ell + \epsilon_\ell) - z_{\ell+1} = \epsilon_\ell$ and $\|\epsilon_\ell\| = o(\lambda^\ell)$. When value divergence is about to occur, the dot product $W_i z_i$ becomes relatively large compared to the scalar $b_i$, and the effect of bias on the output becomes negligible. Therefore, we can recursively deduce that the remaining error term

$$(\lambda - \lambda^{\ell+1}) b_{\ell+1} + \lambda W_\ell \delta_\ell$$

is always negiligle compared with $z'_{\ell+1}$. Hence we know $f_{\theta'}(x)$ is close to $\lambda^L f_\theta(x)$ with only $o(1)$ relative error.

Taking gradient backwards, we know that $\left\|\frac{\partial f}{\partial W_\ell}\right\|$ is proportional to both $\|\tilde{z}_\ell\|$ and $\left\|\frac{\partial f}{\partial z_{\ell+1}}\right\|$. Therefore we know

$$\frac{\partial f_{\theta'}}{\partial W'_\ell} = \tilde{z}'_\ell \frac{\partial f_{\theta'}}{\partial z'_{\ell+1}} = \lambda^\ell \tilde{z}_\ell \cdot \lambda^{L-\ell-1} \frac{\partial f_\theta}{\partial z_{\ell+1}} = \lambda^{L-1} \frac{\partial f_\theta}{\partial W_\ell}.$$

This suggests that all gradients with respect to the weights become scaled by a factor of $\lambda^{L-1}$. The gradients with respect to the biases are proportional to $\lambda^{L-l}$. When $\lambda$ is large enough to render the gradient of the bias term negligible, it follows that $\left\|\nabla_\theta f_{\theta'}(x) - \lambda^{L-1} \nabla_\theta f_\theta(x)\right\| = o(1) \cdot \|\nabla_\theta f_{\theta'}(x)\|$. This equation implies that the gradient updates for the model parameters are dominated by the weights, with negligible contribution from the bias terms. And since NTK is the inner product between gradients, we know $G_{\theta'}(x, x')$'s relative error to $\lambda^{2(L-1)} G_\theta(x, x')$ is $o(1)$ negligible. We run experiments in Appendix F to validate the homogeneity. We leave analysis for Convolution Network [27] and its variants [43; 49; 50] for future work. $\qquad\square$

**Theorem 1** *Suppose that the network's parameter at iteration $t$ is $\theta_t$. For each transition $(s_i, a_i, s_{i+1}, r_i)$ in dataset, denote $r = [r_1, \ldots, r_M]^\top \in \mathbb{R}^M$, $\hat{\pi}_{\theta_t}(s) = \arg\max_a \hat{Q}_{\theta_t}(s, a)$. Denote $x^*_{i,t} = (s_{i+1}, \hat{\pi}_{\theta_t}(s_{i+1}))$. Concatenate all $x^*_{i,t}$ to be $X^*_t$. Denote $u_t = f_{\theta_t}(X) - (r + \gamma \cdot f_{\theta_t}(X^*_t))$ to be TD error vector at iteration $t$. The learning rate $\eta$ is infinitesimal. When the maximal point of $\hat{Q}_{\theta_t}$ is stable as $t$ increases, we have the following evolving equation for $u_{t+1}$*

$$u_{t+1} = (I + \eta A_t) u_t. \tag{3}$$

*where $A = (\gamma \phi_{\theta_t}(X^*_t) - \phi_{\theta_t}(X))^\top \phi_{\theta_t}(X) = \gamma G_{\theta_t}(X^*_t, X) - G_{\theta_t}(X, X)$.*

*Proof.* For the sake of simplicity, denote $Z_t = \nabla_\theta f_\theta(X)\big|_{\theta_t}$, $Z^*_t = \nabla_\theta f_\theta(X^*_t)\big|_{\theta_t}$. The Q-value iteration minimizes loss function $\mathcal{L}$ defined by $\mathcal{L}(\theta) = \frac{1}{2} \|f_\theta(X) - (r + \gamma \cdot f_{\theta_t}(X^*_t))\|_2^2$. Therefore we have the gradient as

$$\frac{\partial \mathcal{L}(\theta)}{\partial \theta} = Z_t (f_\theta(X) - (r + \gamma \cdot f_{\theta_t}(X^*_t))) = Z_t u_t. \tag{4}$$

According to gradient descent, we know $\boldsymbol{\theta}_{t+1} = \boldsymbol{\theta}_t - \eta \boldsymbol{Z}_t \boldsymbol{u}_t$. Since $\eta$ is very small, we know $\boldsymbol{\theta}_{t+1}$ stays within the neighborhood of $\boldsymbol{\theta}_t$. We can Taylor-expand function $f_{\boldsymbol{\theta}}(\cdot)$ near $\boldsymbol{\theta}_t$ as

$$f_{\boldsymbol{\theta}}(\boldsymbol{X}) \approx \nabla_{\boldsymbol{\theta}}^{\top} f_{\boldsymbol{\theta}}(\boldsymbol{X})\Big|_{\boldsymbol{\theta}_t} (\boldsymbol{\theta} - \boldsymbol{\theta}_t) + f_{\boldsymbol{\theta}_t}(\boldsymbol{X}) = \boldsymbol{Z}_t^{\top}(\boldsymbol{\theta} - \boldsymbol{\theta}_t) + f_{\boldsymbol{\theta}_t}(\boldsymbol{X}). \tag{5}$$

$$f_{\boldsymbol{\theta}}(\boldsymbol{X}_t^*) \approx (\boldsymbol{Z}_t^*)^{\top}(\boldsymbol{\theta} - \boldsymbol{\theta}_t) + f_{\boldsymbol{\theta}_t}(\boldsymbol{X}_t^*). \tag{6}$$

When $\eta$ is infinitesimally small, the equation holds. Plug in $\boldsymbol{\theta}_{t+1}$, we know

$$f_{\boldsymbol{\theta}_{t+1}}(\boldsymbol{X}) - f_{\boldsymbol{\theta}_t}(\boldsymbol{X}) = -\eta \boldsymbol{Z}_t^{\top} \boldsymbol{Z}_t \boldsymbol{u}_t = -\eta \cdot \boldsymbol{G}_{\boldsymbol{\theta}_t}(\boldsymbol{X}, \boldsymbol{X}) \boldsymbol{u}_t. \tag{7}$$

$$f_{\boldsymbol{\theta}_{t+1}}(\boldsymbol{X}_t^*) - f_{\boldsymbol{\theta}_t}(\boldsymbol{X}_t^*) = -\eta (\boldsymbol{Z}_t^*)^{\top} \boldsymbol{Z}_t \boldsymbol{u}_t = -\eta \cdot \boldsymbol{G}_{\boldsymbol{\theta}_t}(\boldsymbol{X}_t^*, \boldsymbol{X}) \boldsymbol{u}_t. \tag{8}$$

According to Assumption 2, we know $\|\boldsymbol{X}_t^* - \boldsymbol{X}_{t+1}^*\| = o(\eta)$. So we can substitute $\boldsymbol{X}_{t+1}^*$ into $\boldsymbol{X}_t^*$ since it is high order infinitesimal. Therefore, $\boldsymbol{u}_{t+1}$ boils down to

$$\boldsymbol{u}_{t+1} = f_{\boldsymbol{\theta}_{t+1}}(\boldsymbol{X}) - \boldsymbol{r} - \gamma f_{\boldsymbol{\theta}_{t+1}}(\boldsymbol{X}_{t+1}^*) \tag{9}$$

$$= f_{\boldsymbol{\theta}_t}(\boldsymbol{X}) - \eta \cdot \boldsymbol{G}_{\boldsymbol{\theta}_t}(\boldsymbol{X}, \boldsymbol{X}) \boldsymbol{u}_t - \boldsymbol{r} - \gamma(f_{\boldsymbol{\theta}_t}(\boldsymbol{X}_t^*) - \eta \cdot \boldsymbol{G}_{\boldsymbol{\theta}_t}(\boldsymbol{X}_t^*, \boldsymbol{X}) + o(\eta))\boldsymbol{u}_t \tag{10}$$

$$= \underbrace{f_{\boldsymbol{\theta}_t}(\boldsymbol{X}) - \boldsymbol{r} - \gamma f_{\boldsymbol{\theta}_t}(\boldsymbol{X}_t^*)}_{\boldsymbol{u}_t} + \eta \cdot (\gamma \boldsymbol{G}_{\boldsymbol{\theta}_t}(\boldsymbol{X}_t^*, \boldsymbol{X}) - \boldsymbol{G}_{\boldsymbol{\theta}_t}(\boldsymbol{X}, \boldsymbol{X}) + o(\eta))\boldsymbol{u}_t \tag{11}$$

$$= (\boldsymbol{I} + \eta \boldsymbol{A}_t)\boldsymbol{u}_t. \tag{12}$$

where $\boldsymbol{A} = \gamma \boldsymbol{G}_{\boldsymbol{\theta}_t}(\boldsymbol{X}_t^*, \boldsymbol{X}) - \boldsymbol{G}_{\boldsymbol{\theta}_t}(\boldsymbol{X}, \boldsymbol{X})$. $\square$

**Theorem 3** *Suppose we use SGD optimizer for Q-value iteration with learning rate $\eta$ to be infinitesimal. Given iteration $t > t_0$, and $\boldsymbol{A} = \gamma \bar{\boldsymbol{G}}(\bar{\boldsymbol{X}}^*, \boldsymbol{X}) - \bar{\boldsymbol{G}}(\boldsymbol{X}, \boldsymbol{X})$, where $\bar{\boldsymbol{G}}$ is the Gram matrix under terminal kernel $\bar{k}$. The divergence of $\boldsymbol{u}_t$ is equivalent to whether there exists an eigenvalue $\lambda$ of $\boldsymbol{A}$ such that $\mathrm{Re}(\lambda) > 0$. If converge, we have $\boldsymbol{u}_t = (\boldsymbol{I} + \eta \boldsymbol{A})^{t-t_0} \cdot \boldsymbol{u}_{t_0}$. Otherwise, $\boldsymbol{u}_t$ becomes parallel to the eigenvector of the largest eigenvalue $\lambda$ of $\boldsymbol{A}$, and its norm diverges to infinity at following order*

$$\|\boldsymbol{u}_t\|_2 = O\left(\frac{1}{(1 - C'\lambda\eta t)^{L/(2L-2)}}\right). \tag{13}$$

*for some constant $C'$ to be determined and $L$ is the number of layers of MLP. Specially, when $L = 2$, it reduces to $O\left(\frac{1}{1-C'\lambda\eta t}\right)$.*

*Proof.* According to Assumption 2, max action becomes stable after $t_0$. It implies $\|\boldsymbol{X}_t^* - \bar{\boldsymbol{X}}^*\| = o(1)$. The stability of the NTK direction implies that for some scalar $k_t$ and the specific input $\boldsymbol{X}^*, \boldsymbol{X}$, we have $\boldsymbol{G}_{\boldsymbol{\theta}_t}(\bar{\boldsymbol{X}}^*, \boldsymbol{X}) = k_t \bar{\boldsymbol{G}}(\boldsymbol{X}^*, \boldsymbol{X}) + \boldsymbol{\epsilon}_{t,1}$ and $\boldsymbol{G}_{\boldsymbol{\theta}_t}(\boldsymbol{X}, \boldsymbol{X}) = k_t \bar{\boldsymbol{G}}(\boldsymbol{X}, \boldsymbol{X}) + \boldsymbol{\epsilon}_{t,2}$ where $\|\boldsymbol{\epsilon}_{t,i}\| = o(k_t), i = 1, 2$. Therefore, we have

$$\boldsymbol{A}_t = \gamma \boldsymbol{G}_{\boldsymbol{\theta}_t}(\boldsymbol{X}_t^*, \boldsymbol{X}) - \boldsymbol{G}_{\boldsymbol{\theta}_t}(\boldsymbol{X}, \boldsymbol{X}) \tag{14}$$

$$= \gamma \bar{\boldsymbol{G}}(\boldsymbol{X}_t^*, \boldsymbol{X}) - \bar{\boldsymbol{G}}(\boldsymbol{X}, \boldsymbol{X}) + (\boldsymbol{\epsilon}_{t,1} - \boldsymbol{\epsilon}_{t,2}) \tag{15}$$

$$= k_t(\gamma \bar{\boldsymbol{G}}(\bar{\boldsymbol{X}}^*, \boldsymbol{X}) - \bar{\boldsymbol{G}}(\boldsymbol{X}, \boldsymbol{X})) + (\gamma \boldsymbol{\epsilon}_{t,3} + \boldsymbol{\epsilon}_{t,1} - \boldsymbol{\epsilon}_{t,2}) \tag{16}$$

$$(\boldsymbol{\epsilon}_{t,3} = \bar{\boldsymbol{G}}(\bar{\boldsymbol{X}}_t^*, \boldsymbol{X}) - \bar{\boldsymbol{G}}(\bar{\boldsymbol{X}}^*, \boldsymbol{X}))$$

$$= k_t \boldsymbol{A} + \boldsymbol{\delta}_t \tag{17}$$

$k_t$ equals 1 if the training is convergent, but will float up if the model's predicted Q-value blows up. According to Assumption 2, the rest term $\boldsymbol{\delta}_t = \gamma \boldsymbol{\epsilon}_{t,3} + \boldsymbol{\epsilon}_{t,1} - \boldsymbol{\epsilon}_{t,2}$ is of $o(1)$ norm and thus negligible, we know all the eigenvalues of $\boldsymbol{I} + \eta \boldsymbol{A}$ have form $1 + \eta \lambda_i$. Considering $\eta$ is small enough, we have $|1 + \eta \lambda_i|^2 \approx 1 + 2\eta \mathrm{Re}(\lambda)$. Now suppose if there does not exists eigenvalue $\lambda$ of $\boldsymbol{A}$ satisfies $\mathrm{Re}(\lambda) > 0$, we have $|1 + \eta \lambda_i| \leq 1$. Therefore, the NTK will become perfectly stable so $k_t = 1$ for $t > t_0$, and we have

$$\boldsymbol{u}_t = (\boldsymbol{I} + \eta \boldsymbol{A}_{t-1})\boldsymbol{u}_{t-1} = (\boldsymbol{I} + \eta \boldsymbol{A}_{t-1})(\boldsymbol{I} + \eta \boldsymbol{A}_{t-2})\boldsymbol{u}_{t-2} = \ldots = \prod_{s=t_0}^{t-1}(\boldsymbol{I} + \eta \boldsymbol{A}_s)\boldsymbol{u}_{t_0} \tag{18}$$

$$= \prod_{s=t_0}^{t-1}(\boldsymbol{I} + \eta \boldsymbol{A})\boldsymbol{u}_{t_0} = (\boldsymbol{I} + \eta \boldsymbol{A})^{t-t_0} \cdot \boldsymbol{u}_{t_0}. \tag{19}$$

Otherwise, there exists an eigenvalue for $\boldsymbol{A}$ satisfying $\text{Re}(\lambda) > 0$. Denote the one with the largest real part as $\lambda$, and $\boldsymbol{v}$ to be the corresponding eigenvector. We know matrix $\boldsymbol{I} + \eta\boldsymbol{A}$ also has left eigenvector $\boldsymbol{v}$, whose eigenvalue is $1 + \eta\lambda$. In this situation, we know after each iteration, $\|\boldsymbol{u}_{t+1}\|$ will become larger than $\|\boldsymbol{u}_t\|$. Moreover, to achieve larger and larger prediction values, the model's parameter's norm $\|\boldsymbol{\theta}_t\|$ also starts to explode. We know $\boldsymbol{u}_t$ is homogeneous with respect $\boldsymbol{\theta}_t$ for ReLU networks. The output $f_{\boldsymbol{\theta}_t}(\boldsymbol{X})$ enlarges $p^L$ times when $\boldsymbol{\theta}_t$ enlarges $p$ times. When the reward values is small with respect to the divergent Q-value, TD error $\boldsymbol{u}_t = O(f_{\boldsymbol{\theta}_t}(\boldsymbol{X})) = O(\boldsymbol{\theta}_t^L)$. Besides, according to Lemma1, we know $k_t = O(\|\boldsymbol{\theta}_t\|^{2(L-1)}) = O(\|\boldsymbol{u}_t\|^{2(L-1)/L}) = O(\|\boldsymbol{u}_t\|^{2-2/L})$.

Denote $g(\eta t) = \boldsymbol{v}^\top \boldsymbol{u}_t$, left multiply $\boldsymbol{v}$ to equation $\boldsymbol{u}_{t+1} = (\boldsymbol{I} + \eta k_t \boldsymbol{A})\boldsymbol{u}_t$. we have $g(\eta t + \eta) = (1 + \eta\lambda k_t)g(\eta t)$. Since we know such iteration will let $\boldsymbol{u}_t$ to be dominated by $\boldsymbol{v}$ and align with $\boldsymbol{v}$, we know $g(\eta t) = O(\|\boldsymbol{u}_t\|)$ for large $t$. Therefore $k_t = O(\|\boldsymbol{u}_t\|^{2(L-1)/L}) = C \cdot g(\eta t)^{2-2/L}$. This boils down to $g(\eta t + \eta) = g(\eta t) + C\eta\lambda g(\eta t)^2$, which further becomes

$$\frac{g(\eta t + \eta) - g(\eta t)}{\eta} = C\lambda g(\eta t)^{3-2/L} \tag{20}$$

Let $\eta \to 0$, we have a differential equation $\frac{\mathrm{d}g}{\mathrm{d}t} = C\lambda g(t)^{3-2/L}$. When $L = 1$, the MLP network degenerates to a linear function. The solution of ODE is

$$\|\boldsymbol{u}_t\| = g(\eta t) = C'e^{\lambda t}, \tag{21}$$

reflecting the exponential growth under linear function that has been studied in previous works [47]. When $L > 2$, Solving this ODE gives

$$g(t) = \frac{1}{(1 - C'\lambda t)^{L/(2L-2)}}. \tag{22}$$

So at an infinite limit, we know $\|\boldsymbol{u}_t\| = g(\eta t) = O\left(\frac{1}{(1-C'\lambda\eta t)^{L/(2L-2)}}\right)$. Specially, for the experimental case we study in Figure 2 where $L = 2$, it reduces to $O\left(\frac{1}{1-C'\lambda\eta t}\right)$. We conduct more experiments with $L = 3$ in Appendix C.2 to verify our theoretical findings. $\qquad\square$

It is also worthwhile to point out another important mechanism that we found when the Q-value network is in a steady state of diverging. Although we assume that the learning rate $\eta$ is infinitesimal, the $\arg\max$ function is not $L-$Lipshitz. So why could both NTK function and optimal policy action $\boldsymbol{X}_t^*$ converge in direction? We found that when the model enters such "terminal direction", $(s_i, \hat{\pi}_{\boldsymbol{\theta}_t}(s_i))$ not only becomes the max value state-action pair, but also satisfies

$$\hat{\pi}_{\boldsymbol{\theta}_t}(s_i) = \arg\max_{a\in\mathcal{A}} \sum_{i=1}^{M} k_{\boldsymbol{\theta}_t}(\boldsymbol{x}_a, \boldsymbol{x}_i), \quad \boldsymbol{x}_a = (s_i, a).$$

This means each $\boldsymbol{x}_{t,i}^* = (s_i, \hat{\pi}_{\boldsymbol{\theta}_t}(s_i))$ also gain the maximum value increment among all the possible actions, maintaining its policy action property. Such *stationary policy action for all state* causes the model's parameter to evolve in *static dynamics*, which in turn reassures it to be policy action in the next iteration. Therefore, after the critical point $t_0$, model's parameter evovlves linearly to infinity, its NTK and policy action becomes stationary, and all these factors contribute to the persistence of each other.

## C More Observations and Deduction

### C.1 Model Alignment

In addition to the findings presented in Theorem 1 and Theorem 3, we have noticed several intriguing phenomena. Notably, beyond the critical point, gradients tend to align along a particular direction, leading to an infinite growth of the model's parameters in that same direction. This phenomenon is supported by the observations presented in Figure 16, Figure 17, and Figure 18, where the cosine similarity between the current model parameters and the ones at the ending of training remains close to 1 after reaching a critical point, even as the norm of the parameters continually increases.

## C.2 Terminal Time

Theorem 3 claims $\|\boldsymbol{u}_t\| = O(f_{\boldsymbol{\theta}_t}(\boldsymbol{X})) = O\left(\frac{1}{(1-C'\lambda\eta t)^{L/(2L-2)}}\right)$, implying the relation

$$1/q^{(2L-2)/L} \propto 1 - C'\lambda\eta t. \tag{23}$$

Beisdes, it implies the existence of a "terminal time" $\frac{1}{C'\eta\lambda}$ that the model must crash at a singular point. When the training approaches this singular point, the estimation value and the model's norm explode rapidly in very few steps. We have run an experiment with $L = 2$ in Figure 2, from which we can see that Q-value's inverse proves to decay linearly and eventually becomes Nan at the designated time step. When $L = 3$, from our theretical analysis, we have $1/q^{\frac{4}{3}} \propto 1 - C'\lambda\eta t$. The experimental results in Figure 10 corroborate this theoretical prediction, where the inverse Q-value raised to the power of $4/3$ is proportional to $1 - C'\lambda\eta t$ after a critical point and it eventually reaches a NAN value at the terminal time step.

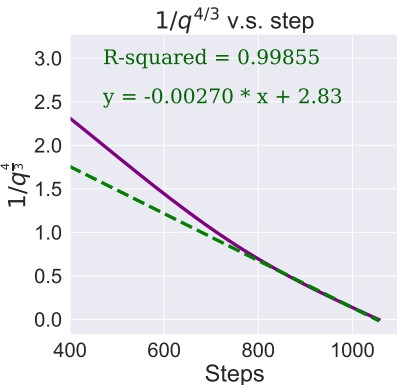

Figure 10: Linear decay with SGD and L=3.

## C.3 Adam Case

In this section, we will prove that if the algorithm employs Adam as the optimizer, the model still suffers divergence. Moreover, we demonstrate that the norm of the network increase linearly, of which the slope is $\eta\sqrt{P}$, where $P$ is the number of parameters and $\eta$ is the learning rate. Also, the Q-value prediction will increase at $L_{th}$-polynomial's rate, where $L$ is the number of layers of model $f_{\boldsymbol{\theta}}$. Experimental results in Figure 4 verified our findings. Besides, we show that all runnings across D4RL environments represents the linear growth of the norm of the Q-network in Figure 16, Figure 17, and Figure 18.

**Theorem 4.** *Suppose we use Adam optimizer for Q-value iteration and all other settings are the same as Theorem 3. After $t > t_0$, the model will diverge if and only if $\lambda_{\max}(\boldsymbol{A}) > 0$. If it diverges, we have $\|\boldsymbol{\theta}_t\| = \eta\sqrt{P}t + o(t)$ and $\|\boldsymbol{u}_t\| = \Theta(t^L)$ where $P$ and $L$ are the number of parameters and the number of layers for network $f_{\boldsymbol{\theta}}$, respectively.*

*Proof.* We only focus on the asymptotic behavior of Adam. So we only care about the dynamics for $t > T$ for some large $T$. Also, at this regime, we know that the gradient has greatly aligned with the model parameters. So we assume that

$$\nabla L(\theta_t) = -C \cdot \theta_t. \quad C > 0 \tag{24}$$

Recall that each iteration of the Adam algorithm has the following steps.

$$g_t = \nabla L(\theta_t), \tag{25}$$

$$m_t = \beta_1 m_{t-1} + (1 - \beta_1)g_t, \tag{26}$$

$$v_t = \beta_2 v_{t-1} + (1 - \beta_2)g_t^2, \tag{27}$$

$$\hat{m}_t = \frac{m_t}{1 - \beta_1^t}, \tag{28}$$

$$\hat{v}_t = \frac{v_t}{1 - \beta_2^t}, \tag{29}$$

$$\theta_{t+1} = \theta_t - \frac{\eta}{\sqrt{\hat{v}_t} + \epsilon}\hat{m}_t. \tag{30}$$

Instead of exactly solving this series, we can verify linear growth is indeed the terminal behavior of $\theta_t$ since we only care about asymptotic order. Assume that $\theta_t = kt$ for $t > T$, we can calculate $m_t$ by dividing both sides of the definition of $m_t$ by $\beta_1^t$, which gives

$$\frac{m_t}{\beta_1^t} = \frac{m_{t-1}}{\beta_1^{t-1}} + \frac{1 - \beta_1}{\beta_1^t}g_t. \tag{31}$$

$$\frac{m_t}{\beta_1^t} = \sum_{s=0}^{t} \frac{1 - \beta_1}{\beta_1^s}g_s. \tag{32}$$

$$m_t = -C\sum_{s=0}^{t}(1 - \beta_1)\beta_1^{t-s}ks = -kCt + o(t) \tag{33}$$

, where $g_t$ is given in Equation (24). Similarly, we have

$$v_t = kC^2t^2 + o(t^2) \tag{34}$$

Hence we verify that

$$\theta_{t+1} - \theta_t = -\eta \cdot \frac{m_t}{1 - \beta_1^t} \cdot \sqrt{\frac{1 - \beta_2^t}{v_t}} \to \eta \cdot \frac{kCt}{\sqrt{k^2C^2t^2}} = \eta$$

therefore we know each iteration will increase each parameter by exactly constant $\eta$. This in turn verified our assumption that parameter $\theta_t$ grows linearly. The slope for the overall parameter is thus $\eta\sqrt{P}$. This can also be verified in Figure 4. When we have $\boldsymbol{\theta}_t = \eta\sqrt{P}\bar{\boldsymbol{\theta}}$, where $\bar{\boldsymbol{\theta}}$ is the normalized parameter, we can further deduce the increasing order of the model's estimation. According to lemma 1, the Q-value estimation (also the training error) increase at speed $O(t^L)$. □

## D LayerNorm's Effect on NTK

In this section, we demonstrate the effect of LayerNorm on SEEM. Our demonstration is just an intuitive explanation rather than a rigorous proof. We show that adding a LayerNorm can effectively reduce the NTK between any $\boldsymbol{x}_0$ and extreme input $\boldsymbol{x}$ down from linear to constant. Since each entry of Gram matrix $\boldsymbol{G}$ is an individual NTK value, we can informally expect that $\boldsymbol{G}(\boldsymbol{X}_t^*, \boldsymbol{X})$'s eigenvalue are greatly reduced when every individual NTK value between any $\boldsymbol{x}_0$ and extreme input $\boldsymbol{x}$ is reduced.

We consider a two-layer MLP. The input is $\boldsymbol{x} \in \mathbb{R}^{d_{in}}$, and the hidden dimension is $d$. The parameters include $\boldsymbol{W} = [\boldsymbol{w}_1, \ldots, \boldsymbol{w}_d]^\top \in \mathbb{R}^{d \times d_{in}}, \boldsymbol{b} \in \mathbb{R}^d$ and $\boldsymbol{a} \in \mathbb{R}^d$. Since for the NTK value, the last layer's bias term has a constant gradient, we do not need to consider it. The forward function of the network is

$$f_{\boldsymbol{\theta}}(\boldsymbol{x}) = \sum_{i=1}^{d} a_i\sigma(\boldsymbol{w}_i^\top\boldsymbol{x} + b_i).$$

**Proposition 1.** *For any input $\boldsymbol{x}$ and network parameter $\boldsymbol{\theta}$, if $\nabla_{\boldsymbol{\theta}}f_{\boldsymbol{\theta}}(\boldsymbol{x}) \neq \boldsymbol{0}$, then we have*

$$\lim_{\lambda \to \infty} k_{\text{NTK}}(\boldsymbol{x}, \lambda\boldsymbol{x}) = \Omega(\lambda) \to \infty. \tag{35}$$

*Proof.* Denote $z_i = \boldsymbol{w}_i^\top \boldsymbol{x} + b_i$, according to condition $\nabla_{\boldsymbol{\theta}} f_{\boldsymbol{\theta}}(\boldsymbol{x}) \neq \boldsymbol{0}$, we know there must exist at least one $i$ such that $z_i > 0$, denote this set as $P$. Now consider all the $i \in [d]$ that satisfy $z_i > 0$ and $\boldsymbol{w}_i^\top \boldsymbol{x} > 0$ (otherwise take opposite sign of $\lambda$), we have

$$\frac{\partial f}{\partial a_i}\bigg|_{\boldsymbol{x}} = \sigma(\boldsymbol{w}_i^\top \boldsymbol{x} + b_i) = \boldsymbol{w}_i^\top \boldsymbol{x} + b_i, \tag{36}$$

$$\frac{\partial f}{\partial \boldsymbol{w}_i}\bigg|_{\boldsymbol{x}} = a_i \boldsymbol{x}, \tag{37}$$

$$\frac{\partial f}{\partial b_i}\bigg|_{\boldsymbol{x}} = a_i. \tag{38}$$

Similarly, we have

$$\frac{\partial f}{\partial a_i}\bigg|_{\lambda \boldsymbol{x}} = \sigma(\lambda \boldsymbol{w}_i^\top \boldsymbol{x} + b_i) = \lambda \boldsymbol{w}_i^\top \boldsymbol{x} + b_i, \tag{39}$$

$$\frac{\partial f}{\partial \boldsymbol{w}_i}\bigg|_{\lambda \boldsymbol{x}} = \lambda a_i \boldsymbol{x}, \tag{40}$$

$$\frac{\partial f}{\partial b_i}\bigg|_{\lambda \boldsymbol{x}} = a_i. \tag{41}$$

So we have

$$\sum_{i \in P} \left\langle \frac{\partial f(\boldsymbol{x})}{\partial \boldsymbol{\theta}_i}, \frac{\partial f(\lambda \boldsymbol{x})}{\partial \boldsymbol{\theta}_i} \right\rangle = \lambda \left( (\boldsymbol{w}_i^\top \boldsymbol{x})^2 + b_i \boldsymbol{w}_i^\top \boldsymbol{x} + a_i^2 \|\boldsymbol{x}\|^2 \right) + O(1) = \Theta(\lambda).$$

Denote $N = \{1, \dots, d\} \setminus P$. We know for every $j \in N$ either $\frac{\partial f(\boldsymbol{x})}{\partial a_j} = \frac{\partial f(\boldsymbol{x})}{\partial b_j} = \frac{\partial f(\boldsymbol{x})}{\partial \boldsymbol{w}_j} = 0$, or $\boldsymbol{w}_j^\top \boldsymbol{x} < 0$. For the latter case, we know $\lim_{\lambda \to \infty} \frac{\partial f(\lambda \boldsymbol{x})}{\partial a_j} = \frac{\partial f(\lambda \boldsymbol{x})}{\partial b_j} = \frac{\partial f(\lambda \boldsymbol{x})}{\partial \boldsymbol{w}_j} = 0$. In both cases, we have

$$\lim_{\lambda \to \infty} \sum_{j \in N} \left\langle \frac{\partial f(\boldsymbol{x})}{\partial \boldsymbol{\theta}_j}, \frac{\partial f(\lambda \boldsymbol{x})}{\partial \boldsymbol{\theta}_j} \right\rangle = 0.$$

Therefore, according to the definition of NTK, we have

$$\lim_{\lambda \to \infty} k_{\text{NTK}}(\boldsymbol{x}, \lambda \boldsymbol{x}) = \lim_{\lambda \to \infty} \left\langle \frac{\partial f(\boldsymbol{x})}{\partial \boldsymbol{\theta}_i}, \frac{\partial f(\lambda \boldsymbol{x})}{\partial \boldsymbol{\theta}_i} \right\rangle = \Theta(\lambda) \to \infty.$$

$\square$

For the model equipped with LayerNorm, the forward function becomes

$$f_{\boldsymbol{\theta}}(\boldsymbol{x}) = \boldsymbol{a}^\top \sigma(\psi(\boldsymbol{W}\boldsymbol{x} + \boldsymbol{b})),$$

where $\psi(\cdot)$ is the layer normalization function defined as

$$\psi(\boldsymbol{x}) = \sqrt{d} \cdot \frac{\boldsymbol{x} - \mathbf{1}\mathbf{1}^\top \boldsymbol{x}/d}{\|\boldsymbol{x} - \mathbf{1}\mathbf{1}^\top \boldsymbol{x}/d\|}.$$

Denote $\boldsymbol{P} = \boldsymbol{I} - \mathbf{1}\mathbf{1}^\top/d$, note that the derivative of $\psi(\cdot)$ is

$$\dot{\psi}(\boldsymbol{x}) = \frac{\partial \psi(\boldsymbol{x})}{\partial \boldsymbol{x}} = \sqrt{d} \cdot \left( \frac{\boldsymbol{I}}{\|\boldsymbol{P}\boldsymbol{x}\|} - \frac{\boldsymbol{P}\boldsymbol{x}\boldsymbol{x}^\top \boldsymbol{P}}{\|\boldsymbol{P}\boldsymbol{x}\|^3} \right) \boldsymbol{P}. \tag{42}$$

Specially, we have

$$\psi(\lambda \boldsymbol{x}) = \sqrt{d} \cdot \frac{\lambda \boldsymbol{x} - \lambda \mathbf{1}\mathbf{1}^\top \boldsymbol{x}/d}{\lambda \|\boldsymbol{x} - \mathbf{1}\mathbf{1}^\top \boldsymbol{x}/d\|} = \psi(\boldsymbol{x}). \tag{43}$$

Now we state the second proposition.

**Proposition 2.** *For any input $\boldsymbol{x}$ and network parameter $\boldsymbol{\theta}$ and any direction $\boldsymbol{v} \in \mathbb{R}^{d_{in}}$, if the network has LayerNorm, then we know there exists a universal constant $C$, such that for any $\lambda \geq 0$, we have*

$$k_{\text{NTK}}(\boldsymbol{x}, \boldsymbol{x} + \lambda \boldsymbol{v}) \leq C. \tag{44}$$

*Proof.* Since for finite range, there always exists a constant upper bound, we just need to analyze the case for $\lambda \to +\infty$ and shows that it is constant bounded. First compute $\nabla_{\boldsymbol{\theta}} f_{\boldsymbol{\theta}}(\boldsymbol{x})$ and get

$$\left.\frac{\partial f}{\partial \boldsymbol{a}}\right|_{\boldsymbol{x}} = \sigma(\psi(\boldsymbol{W}\boldsymbol{x} + \boldsymbol{b})), \tag{45}$$

$$\left.\frac{\partial f}{\partial \boldsymbol{W}}\right|_{\boldsymbol{x}} = \boldsymbol{a}^\top \sigma'(\psi(\boldsymbol{W}\boldsymbol{x} + \boldsymbol{b}))\dot{\psi}(\boldsymbol{W}\boldsymbol{x} + \boldsymbol{b})\boldsymbol{x}, \tag{46}$$

$$\left.\frac{\partial f}{\partial \boldsymbol{b}}\right|_{\boldsymbol{x}} = \boldsymbol{a}^\top \sigma'(\psi(\boldsymbol{W}\boldsymbol{x} + \boldsymbol{b}))\dot{\psi}(\boldsymbol{W}\boldsymbol{x} + \boldsymbol{b}). \tag{47}$$

These quantities are all constant bounded. Next we compute $\lim_{\lambda \to \infty} \nabla_{\boldsymbol{\theta}} f_{\boldsymbol{\theta}}(\boldsymbol{x} + \lambda \boldsymbol{v})$

$$\left.\frac{\partial f}{\partial \boldsymbol{a}}\right|_{\boldsymbol{x}+\lambda\boldsymbol{v}} = \sigma(\psi(\boldsymbol{W}(\boldsymbol{x} + \lambda\boldsymbol{v}) + \boldsymbol{b}))), \tag{48}$$

$$\left.\frac{\partial f}{\partial \boldsymbol{W}}\right|_{\boldsymbol{x}+\lambda\boldsymbol{v}} = \boldsymbol{a}^\top \sigma'(\psi(\boldsymbol{W}(\boldsymbol{x} + \lambda\boldsymbol{v}) + \boldsymbol{b})))\dot{\psi}(\boldsymbol{W}(\boldsymbol{x} + \lambda\boldsymbol{v}) + \boldsymbol{b})(\boldsymbol{x} + \lambda\boldsymbol{v}), \tag{49}$$

$$\left.\frac{\partial f}{\partial \boldsymbol{b}}\right|_{\boldsymbol{x}+\lambda\boldsymbol{v}} = \boldsymbol{a}^\top \sigma'(\psi(\boldsymbol{W}\boldsymbol{x} + \boldsymbol{b}))\dot{\psi}(\boldsymbol{W}(\boldsymbol{x} + \lambda\boldsymbol{v}) + \boldsymbol{b}). \tag{50}$$

According to the property of LayerNorm in Equation (43), we have

$$\overline{\lim_{\lambda\to\infty}}\left.\frac{\partial f}{\partial \boldsymbol{a}}\right|_{\boldsymbol{x}+\lambda\boldsymbol{v}} = \overline{\lim_{\lambda\to\infty}} \sigma(\psi(\boldsymbol{W}(\boldsymbol{x} + \lambda\boldsymbol{v}) + \boldsymbol{b})) \tag{51}$$

$$= \sigma(\psi(\boldsymbol{W}(\lambda\boldsymbol{v}))) \tag{52}$$

$$= \sigma(\psi(\boldsymbol{W}\boldsymbol{v})) = \text{Constant} \tag{53}$$

$$\overline{\lim_{\lambda\to\infty}}\left.\frac{\partial f}{\partial \boldsymbol{W}}\right|_{\boldsymbol{x}+\lambda\boldsymbol{v}} = \overline{\lim_{\lambda\to\infty}} \boldsymbol{a}^\top \sigma'(\psi(\boldsymbol{W}(\boldsymbol{x} + \lambda\boldsymbol{v}) + \boldsymbol{b})))\dot{\psi}(\boldsymbol{W}(\boldsymbol{x} + \lambda\boldsymbol{v}) + \boldsymbol{b})(\boldsymbol{x} + \lambda\boldsymbol{v}) \tag{54}$$

$$= \overline{\lim_{\lambda\to\infty}} \boldsymbol{a}^\top \sigma'(\psi(\boldsymbol{W}\boldsymbol{v})))\dot{\psi}(\boldsymbol{W}(\boldsymbol{x} + \lambda\boldsymbol{v}) + \boldsymbol{b})(\boldsymbol{x} + \lambda\boldsymbol{v}) \tag{55}$$

$$= \overline{\lim_{\lambda\to\infty}} \boldsymbol{a}^\top \sigma'(\psi(\boldsymbol{W}\boldsymbol{v})))\sqrt{d} \cdot \left(\frac{\boldsymbol{I}}{\|\boldsymbol{P}\lambda\boldsymbol{W}\boldsymbol{v}\|} - \frac{\boldsymbol{P}(\lambda\boldsymbol{W}\boldsymbol{v})(\lambda\boldsymbol{W}\boldsymbol{v})^\top\boldsymbol{P}}{\|\boldsymbol{P}(\lambda\boldsymbol{W}\boldsymbol{v})\|^3}\right)\boldsymbol{P}(\boldsymbol{x} + \lambda\boldsymbol{v}) \tag{56}$$

$$= \overline{\lim_{\lambda\to\infty}} \boldsymbol{a}^\top \sigma'(\psi(\boldsymbol{W}\boldsymbol{v})))\sqrt{d} \cdot \left(\frac{\boldsymbol{P}(\boldsymbol{x} + \lambda\boldsymbol{v})}{\lambda\|\boldsymbol{P}\boldsymbol{W}\boldsymbol{v}\|} - \frac{\boldsymbol{P}\boldsymbol{W}\boldsymbol{v}\boldsymbol{v}^\top\boldsymbol{W}^\top\boldsymbol{P}(\boldsymbol{x} + \lambda\boldsymbol{v})}{\lambda\|\boldsymbol{P}\boldsymbol{W}\boldsymbol{v}\|^3}\right) \tag{57}$$

$$= \boldsymbol{a}^\top \sigma'(\psi(\boldsymbol{W}\boldsymbol{v})))\sqrt{d} \cdot \left(\frac{\boldsymbol{P}\boldsymbol{v}}{\|\boldsymbol{P}\boldsymbol{W}\boldsymbol{v}\|} - \frac{\boldsymbol{P}\boldsymbol{W}\boldsymbol{v}\boldsymbol{v}^\top\boldsymbol{W}^\top\boldsymbol{P}\boldsymbol{v}}{\|\boldsymbol{P}\boldsymbol{W}\boldsymbol{v}\|^3}\right) \tag{58}$$

$$= \text{Constant}. \tag{59}$$

$$\overline{\lim_{\lambda\to\infty}}\left.\frac{\partial f}{\partial \boldsymbol{b}}\right|_{\boldsymbol{x}+\lambda\boldsymbol{v}} = \overline{\lim_{\lambda\to\infty}} \boldsymbol{a}^\top \sigma'(\psi(\boldsymbol{W}\boldsymbol{v})))\sqrt{d} \cdot \left(\frac{\boldsymbol{I}}{\lambda\|\boldsymbol{P}\boldsymbol{W}\boldsymbol{v}\|} - \frac{\boldsymbol{P}\boldsymbol{W}\boldsymbol{v}\boldsymbol{W}\boldsymbol{v})^\top\boldsymbol{P}}{\lambda\|\boldsymbol{P}(\boldsymbol{W}\boldsymbol{v})\|^3}\right)\boldsymbol{P} \tag{60}$$

$$= 0. \tag{61}$$

Therefore we know its limit is also constant bounded. So we know there exists a universal constant with respect to $\boldsymbol{\theta}, \boldsymbol{x}, \boldsymbol{v}$ such that $k_{\text{NTK}}(\boldsymbol{x}, \boldsymbol{x} + \lambda\boldsymbol{v}) = \left\langle \frac{\partial f(\boldsymbol{x})}{\partial \boldsymbol{\theta}_i}, \frac{\partial f(\boldsymbol{x}+\lambda\boldsymbol{v})}{\partial \boldsymbol{\theta}_i} \right\rangle \leq C$.

## E  Experiment Setup

**SEEM Experiments**  For the experiments presented in Section Section 3.1, we adopted TD3 as our baseline, but with a modification: instead of using an exponential moving average (EMA), we directly copied the current Q-network as the target network. The Adam optimizer was used with a learning rate of 0.0003, $\beta_1 = 0.9$, and $\beta_2 = 0.999$. The discount factor, $\gamma$, was set to 0.99. Our code builds upon the existing TD3+BC framework, which can be found at https://github.com/sfujim/TD3_BC.

**SEEM Reduction Experiments** For the experiments discussed in Section Section 4, we maintained the same configuration as in the SEEM experiments, with the exception of adding regularizations and normalizations. LayerNorm was implemented between the linear and activation layers with learnable affine parameters, applied to all hidden layers excluding the output layer. WeightNorm was applied to the output layer weights.

**Offline RL Algorithm Experiments** For the X% Mujoco Locomotion offline dataset experiments presented in Section Section 5, we used true offline RL algorithms including TD3+BC, IQL, Diff-QL, and CQL as baselines. We implement our method on the top of official implementations of TD3+BC and IQL; for CQL, we use a reliable JAX implementation at https://github.com/sail-sg/offbench [25]; for Diffusion Q-learning (*i.e.*, combining TD3+BC and diffusion policy), we use an efficient JAX implementation EDP [24]. LayerNorm was directly added to the critic network in these experiments. For the standard Antmaze experiment, we assign a reward +10 to the successful transitions, otherwise 0. Considering the extreme suboptimality in Antmaze medium and large datasets, we adopt Return-based Offline Prioritized Replay (OPER-R) with $p_{\text{base}} = 0$ to rebalance these suboptimal datasets [58; 59; 18]. Specifically, instead of uniform sampling during policy improvement, we assign a higher probability to the transitions in the successful trajectories. Meanwhile, a uniform sampler is still used for policy evaluation. When removing OPER-R, diff-QL with LayerNorm can still achieve a stable performance with slightly lower scores (see Table 2).

**Linear Decay of Inverse Q-value with SGD** Given that the explosion in D4RL environments occurs very quickly in the order of $\frac{1}{1-C'\lambda\eta t}$ and is difficult to capture, we opted to use a simple toy task for these experiments. The task includes a continuous two-dimensional state space $s = (x_1, x_2) \in \mathcal{S} = R^2$, where the agent can freely navigate the plane. The action space is discrete, with 8 possible actions representing combinations of forward or backward movement in two directions. Each action changes the state by a value of 0.01. All rewards are set to zero, meaning that the true Q-value should be zero for all state-action pairs. For this task, we randomly sampled 100 state-action pairs as our offline dataset. The Q-network was implemented as a two-layer MLP with a hidden size of 200. We used SGD with a learning rate of 0.01, and the discount factor, $\gamma$ was set to 0.99.

Table 2: Ablate on Antmaze. The best score metrics is presented in the parenthesis.

| Dataset | diff-QL | ours w.o. OPER-R | ours |
|---|---|---|---|
| antmaze-umaze-v0 | 95.6 (96.0) | 94.3 | $94.3 \pm 0.5$ (97.0) |
| antmaze-umaze-diverse-v0 | (84.0) | 88.5 | $\mathbf{88.5 \pm 6.1}$ (95.0) |
| antmaze-medium-play-v0 | 0.0 (79.8) | 77.0 | $85.6 \pm 1.7$ (92.0) |
| antmaze-medium-diverse-v0 | 6.4 (82.0) | 70.4 | $83.9 \pm 1.6$ (90.7) |
| antmaze-large-play-v0 | 1.6 (49.0) | 47.8 | $\mathbf{65.4 \pm 8.6}$ (74.0) |
| antmaze-large-diverse-v0 | 4.4 (61.7) | 60.3 | $\mathbf{67.1 \pm 1.8}$ (75.7) |
| average | 29.6 (75.4) | 73.1 | $\mathbf{80.8}$ (87.4) |

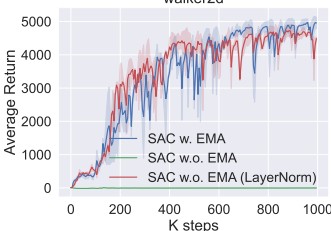
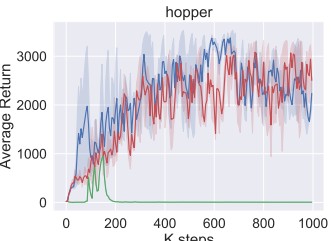

Figure 11: SAC in the online setting.

# F More Experiments

## F.1 Validate the homogeneity in Lemma 1.

We define a family of 3-layer ReLU-activated MLPs (with bias term) with different scaling $\lambda$ of the same network parameter $\theta$ (from D4RL training checkpoints when value divergence happens), and

feed these networks with the same input. We show one example in the below table, which confirms that the $L$-degree homogeneity of output, gradient and NTK is valid at high precision, the NTK is almost parallel, too. The larger the scaling factor is, the more accurate $k^L$ increasing pattern is for output ( $k^{L-1}$ for gradient and $k^{2(L-1)}$ for NTK). Further, we empirically validated the $L$-degree homogeneity of NTK holds for all checkpoints in D4RL experiments where divergence happens.

Table 3: Validate homogeneity in Lemma 1.

| $\lambda$ | 5 | 10 |
|---|---|---|
| $f_{\lambda\theta}(x)/f_\theta(x)$ | 124.99 ($5^3$) | 999.78 ($10^3$) |
| grad scale | 25.001 ($5^2$) | 100.001 ($10^2$) |
| NTK scale | 624.99($5^4$) | 9999.84 ($10^4$) |
| NTK cos | $1-10^{-6}$ | $1-10^{-6}$ |

## F.2  Ablate the contributions of EMA, Double-Q, and LayerNorm.

We conducted an ablation study in the following sequence: "DDPG without EMA", "DDPG + EMA", "DDPG + EMA + DoubleQ", and "DDPG + EMA + DoubleQ + LayerNorm". Each successive step built upon the last. We carried out these experiments in settings of 10% and 50% Mujoco - environments notably susceptible to divergence. We report the total scores across nine tasks in Figure 12. It clearly reveals that the EMA and double-Q, which are used as common pracitce in popular offline RL algorithms such as CQL, TD3+BC contribute to preventing divergence and thus boosting performance. Nevertheless, built upon EMA and DoubleQ, our LayerNorm can futher significantly boost the performance by surpressing divergence.

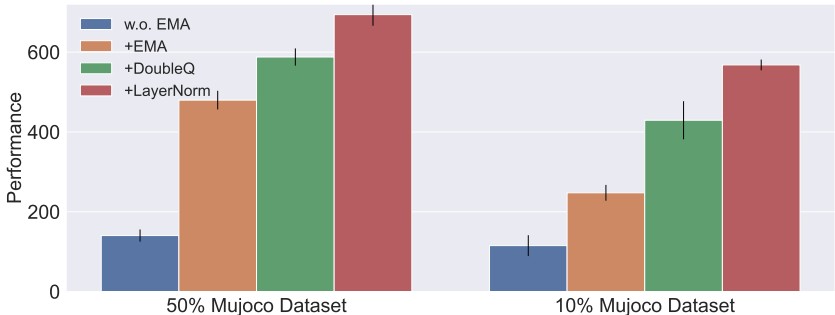

Figure 12: Ablate EMA, Double-Q, and LayerNorm on 10% and 50% Mujoco datasets.

## F.3  Benchmarking Normalizations.

Previously, we have demonstrated that LayerNorm, BatchNorm, and WeightNorm can effectively maintain a low SEEM and stabilize Q convergence in Section 4. Our next goal is to identify the most suitable regularization method for the value network in offline RL. Prior research has shown that divergence is correlated with poor control performance[48; 19]. In this context, we evaluate the effectiveness of various regularization techniques based on their performance in two distinct settings - the Antmaze task and the X% Mujoco dataset we mentioned above. Previous offline RL algorithms have not performed particularly well in these challenging scenarios. As displayed in Figure 13, TD3+BC, when coupled with layer normalization or batch normalization, yields significant performance enhancement on the 10% Mujoco datasets. The inability of batch normalization to improve the performance might be attributed to the oscillation issue previously discussed in Section 4. In the case of Antmaze tasks, which contain numerous suboptimal trajectories, we select TD3 with a diffusion policy, namely Diff-QL [51], as our baseline. The diffusion policy is capable of capturing multi-modal behavior. As demonstrated in Figure 13 and Table 4, LayerNorm can markedly enhance performance on challenging Antmaze tasks. In summary, we empirically find LayerNorm to be a suitable normalization for the critic in offline RL.

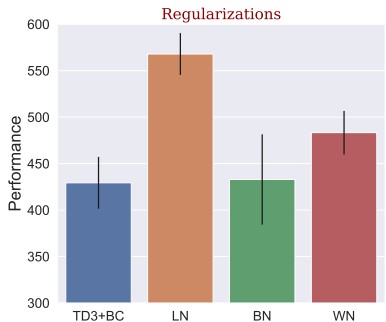

Figure 13: Normalizations effect on 10% Mujoco Locomotion Datasets.

Table 4: Normalizations effect on two challenging Antmaze tasks.

| Dataset | diff-QL | LN | BN | WN |
|---|---|---|---|---|
| antmaze-large-play-v0 | 1.6 | **72.7** | 1.0 | **35.0** |
| antmaze-large-diverse-v0 | 4.4 | **66.5** | 2.1 | **42.5** |

## F.4 How LayerNorm should be added.

The inclusion of LayerNorm is situated between the linear and activation layers. However, the ideal configuration for adding LayerNorm can vary and may depend on factors such as 1) the specific layers to which LayerNorm should be added, and 2) whether or not to apply learnable per-element affine parameters. To explore these variables, we conducted an assessment of their impacts on performance in the two most challenging Antmaze environments. Our experimental setup mirrored that of the Antmaze experiments mentioned above, utilizing a three-layer MLP critic with a hidden size configuration of (256,256,256). We evaluated variants where LayerNorm was only applied to a portion of hidden layers and where learnable affine parameters were disabled. As seen in Table 5, the performances with LayerNorm applied solely to the initial layers LN (0), LN (0,1) are considerably lower compared to the other setups in the 'antmaze-large-play-v0' task, while applying LayerNorm to all layers LN(0,1,2) seems to yield the best performance. For the 'antmaze-large-diverse-v0' task, performances seem to be more consistent across different LayerNorm applications. Overall, this analysis suggests that applying LayerNorm to all layers tends to yield the best performance in these tasks. Also, the utilization of learnable affine parameters appears less critical in this context.

Table 5: The effect of LayerNorm implementations on two challenging Antmaze tasks.

| Dataset | w.o. LN | LN (0) | LN (0,1,) | LN (1,2) | LN (2) | LN (0,1,2) | LN (no learnable) |
|---|---|---|---|---|---|---|---|
| antmaze-large-play-v0 | 1.6 | 0 | 0 | 8.3 | 17.8 | **72.7** | **72.8** |
| antmaze-large-diverse-v0 | 4.4 | **60.2** | **68** | **77.1** | 65.5 | 66.5 | 66.7 |

## F.5 Baird's Counterexample.

Here, we will present our solution allow linear approximation to solve Baird's Counterexample , which was first used to show the potential divergence of Q-learning. Baird's Counterexample specifically targets linear approximation. Neural networks exhibited stable convergence within the Baird counterexample scenario. Thus, we investigated cases of divergence within linear situations and discovered that incorporating LayerNorm in front of linear approximation could effectively lead to convergence (see Figure 14). This finding is not entirely surprising. Since a linear regression model can be considered a special case of neural networks, our analysis can be directly applied to linear settings.

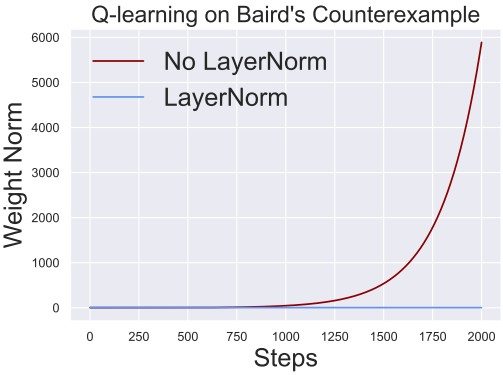

Figure 14: Baird's Counterexample.

## G Discussion

**SEEM and Deadly Triad.** Deadly Triad is a term that refers to a problematic interaction observed in reinforcement learning algorithms, where off-policy learning, function approximation, and bootstrapping converge, leading to divergence during training. Existing studies primarily analyze linear functions as Q-values, which tend to limit the analysis to specific toy examples. In contrast, our work uses NTK theory to provide an in-depth understanding of the divergence of Q-values in non-linear neural networks in realistic settings, and introduces SEEM as a tool to depict such divergence. SEEM can be used to understand the Deadly Triad as follows: If a policy is nearly on-policy, $\boldsymbol{X}_t^*$ is merely a perturbation of $\boldsymbol{X}$. Consequently, $\boldsymbol{A}_t = \gamma \boldsymbol{G}_{\boldsymbol{\theta}_t}(\boldsymbol{X}_t^*, \boldsymbol{X}) - \boldsymbol{G}_{\boldsymbol{\theta}_t}(\boldsymbol{X}, \boldsymbol{X}) \approx (\gamma - 1)\boldsymbol{G}_{\boldsymbol{\theta}_t}(\boldsymbol{X}, \boldsymbol{X})$, with $\boldsymbol{G}$ tending to be negative-definite. Without function approximation, the update of $Q(\boldsymbol{X})$ will not influence $Q(\boldsymbol{X}_t^*)$, and the first term in $\boldsymbol{A}_t$ becomes zero. $\boldsymbol{A}_t = -\boldsymbol{G}_{\boldsymbol{\theta}_t}(\boldsymbol{X}, \boldsymbol{X})$ ensures that SEEM is non-positive and Q-value iteration remains non-expansive. If we avoid bootstrapping, the value iteration transforms into a supervised learning problem with well-understood convergence properties. However, when all three components in Deadly Triad are present, the NTK analysis gives rise to the form $\boldsymbol{A}_t = \gamma \boldsymbol{G}_{\boldsymbol{\theta}_t}(\boldsymbol{X}_t^*, \boldsymbol{X}) - \boldsymbol{G}_{\boldsymbol{\theta}_t}(\boldsymbol{X}, \boldsymbol{X})$, which may result in divergence if the SEEM is positive.

**Linearity Prediction Outside Dataset Range.** As has been pointed out in [54] (Fig. 1), ReLU-activated MLP without any norm layer becomes a linear function for the points that are excessively out of the dataset range. This fact can be comprehended in the following way: Consider an input $\lambda x$, the activation state for every neuron in the network becomes deterministic when $\lambda \to \infty$. Therefore, the ReLU activation layer becomes a multiplication of a constant 0-1 mask, and the whole network degenerates to a linear function. If the network becomes a linear function for inputs $\lambda x_0, \lambda \geq C$, this will cause an arbitrarily large NTK value between extreme points $\lambda x_0$ and dataset point $x_0$. Because now $f_\theta(\lambda x)$ can be written in equivalent form $W^T(\lambda x)$, so $\phi(\lambda x) = \lambda W$ becomes linear proportional to $\lambda$. This further induces linear NTK value between $x$ and $\lambda x$ that is unbounded. This phenomenon can be intuitively interpreted as below, since the value prediction of $f_\theta(x)$ on line $\lambda x_0$ is almost a linear function $W^T x$ for large $\lambda$, any subtle change of effective parameter $W$ will induce a large change on $\lambda x_0$ for large $\lambda$. This demonstrates a strong correlation between the value prediction between point $x_0$ and far-away extreme point $\lambda x_0$. The rigorous proof can be found in Proposition 1.

**Policy Constraint and LayerNorm.** We have established a connection between SEEM and value divergence. As shown in Figure 6, policy constraint alone can also control SEEM and prevent divergence. In effect, policy constraint addresses an aspect of the Deadly Triad by managing the degree of off-policy learning. However, an overemphasis on policy constraint, leading to excessive bias, can be detrimental to the policy and impair performance, as depicted in Figure 7. Building on this insight, we focus on an orthogonal perspective in deadly triad - regularizing the generalization capacity of the critic network. Specifically, we propose the use of LayerNorm in the critic network to inhibit value divergence and enhance agent performance. Policy constraint introduces an explicit bias into the policy, while LayerNorm does not. Learning useful information often requires some degree of prior bias towards offline dataset, but too much can hinder performance. LayerNorm, offering an orthogonal perspective to policy constraint, aids in striking a better balance.

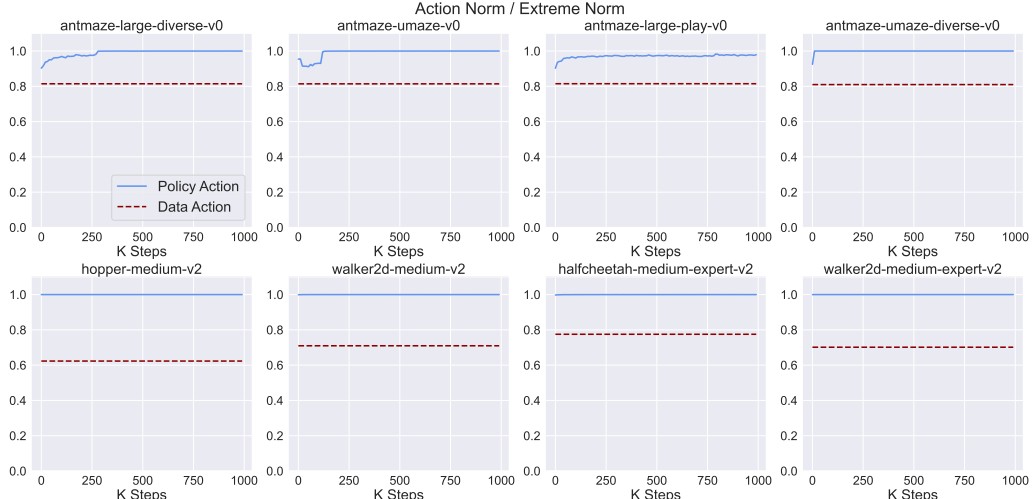

Figure 15: An observation that policy actions tend to gravitate toward extreme points when Q-value divergence occurs. In D4RL environments, where action spaces are bounded, the ratio of the policy action norm to the norm of points at the boundary (i.e., extreme points) provides an indication of whether the current action is approaching these extreme points. To facilitate the observation of Q-value divergence, the experimental setup employed is consistent with that described in Section 3 of the paper. From the figure, it can be seen that the ratios for policy actions are nearly 1, suggesting a close proximity to the extreme points. In contrast, the ratios for actions within the dataset are significantly lower than 1.

## H   More Visualization Results

In  Assumption 2, we posit that the direction of NTK and the policy remains stable following a certain period of training. We validates this assumption through experimental studies. We observe the convergence of the NTK trajectory and policy in all D4RL Mujoco Locomotion and Antmaze tasks, as depicted in the first two columns of Figures Figure 16,  Figure 17, and  Figure 18. We also illustrate the linear growth characteristic of Adam optimization (as outlined in Theorem Theorem 4) in the fourth column. As a consequence, the model parameter vectors maintain a parallel trajectory, keeping the cosine similarity near 1 as shown in the third column. Figure 19 and  Figure 20 showcase how SEEM serves as a "divergence detector"in Mujoco and Antamze tasks. The surge in the SEEM value is consistently synchronized with an increase in the estimated Q-value.

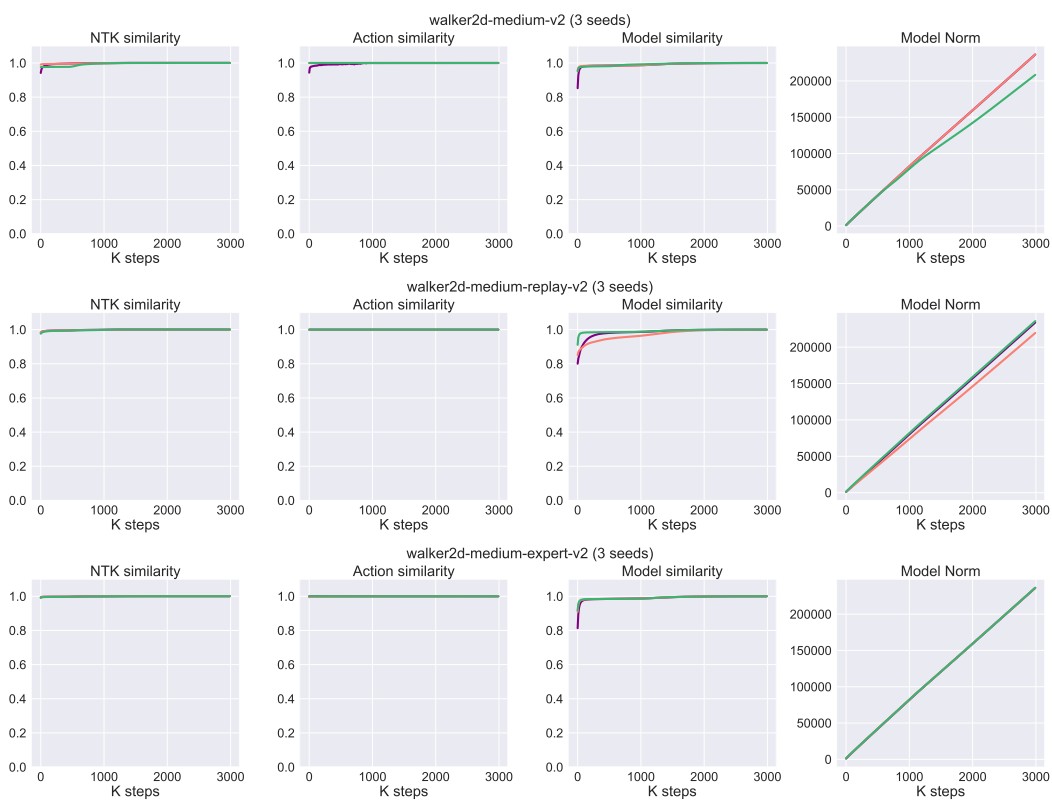

Figure 16: NTK similarity, action similarity, model parameter similarity, and model parameter norm curves in D4RL Mujoco Walker2d tasks.

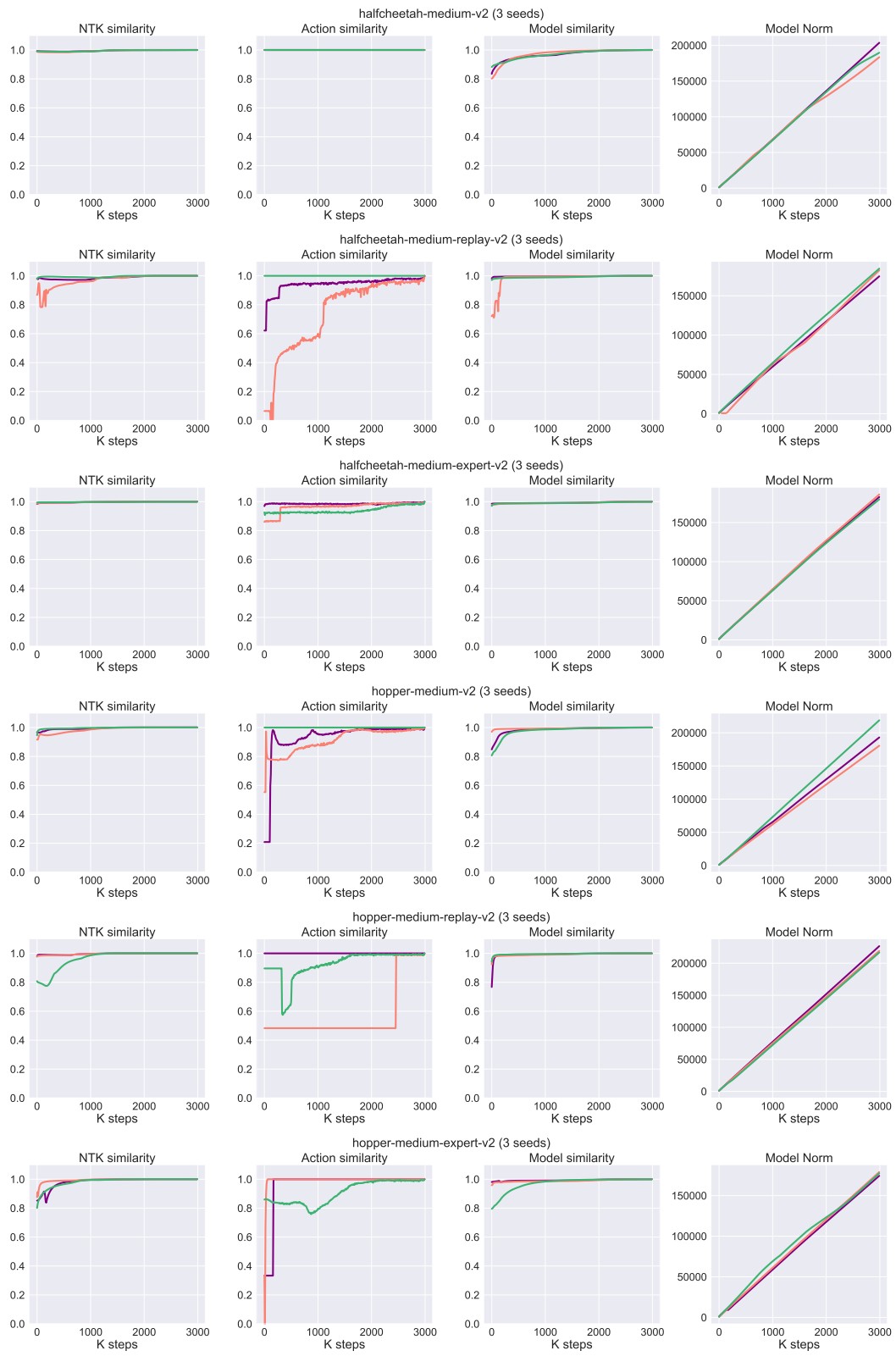

Figure 17: NTK similarity, action similarity, model parameter similarity, and model parameter norm curves in D4RL Mujoco Halfcheetah and Hopper tasks.

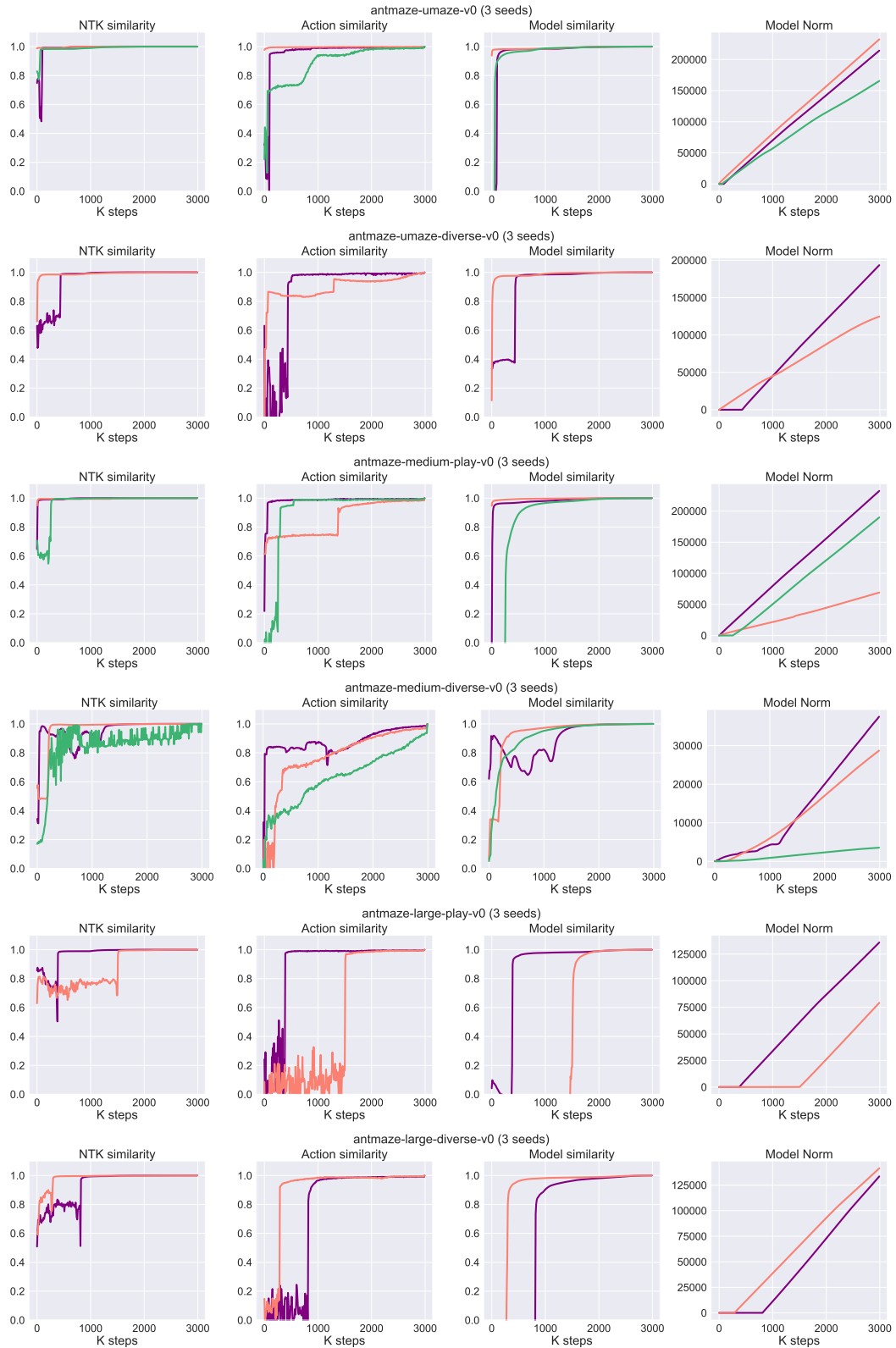

Figure 18: NTK similarity, action similarity, model parameter similarity, and model parameter norm curves in Antmaze tasks.

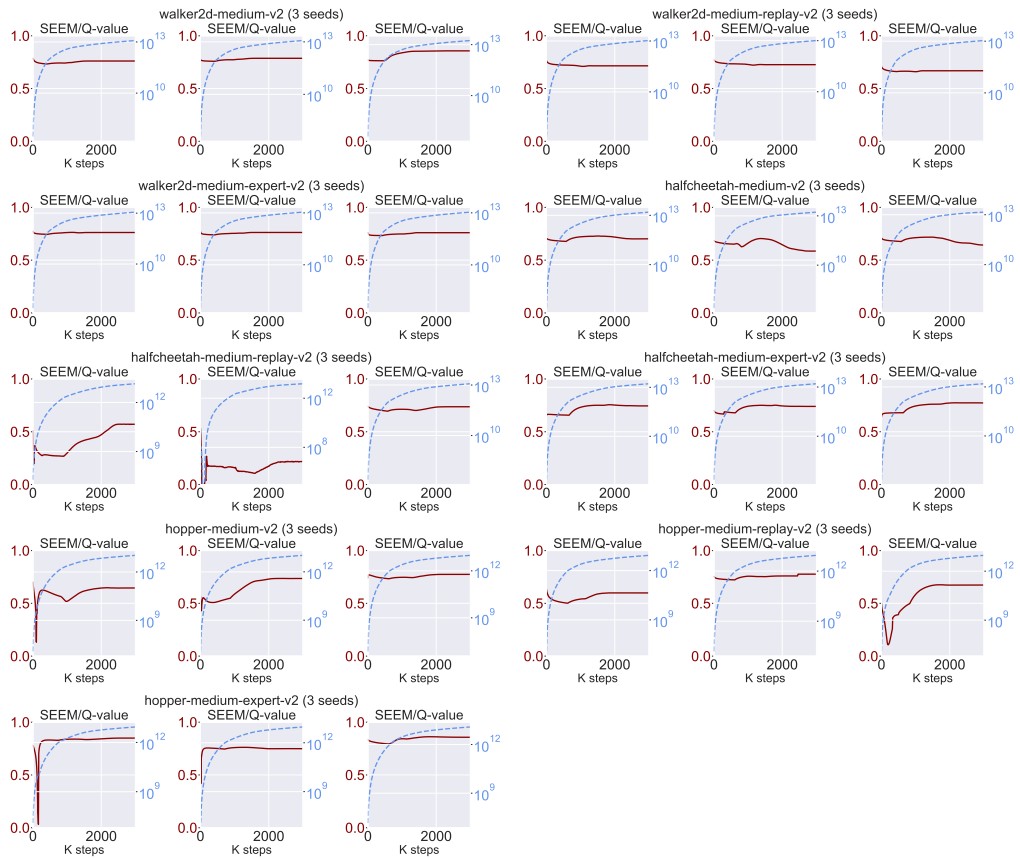

Figure 19: The normalized kernel matrix's SEEM (in red) and the estimated Q-value (in blue) in D4RL Mujoco tasks. For each environment, results from three distinct seeds are reported.

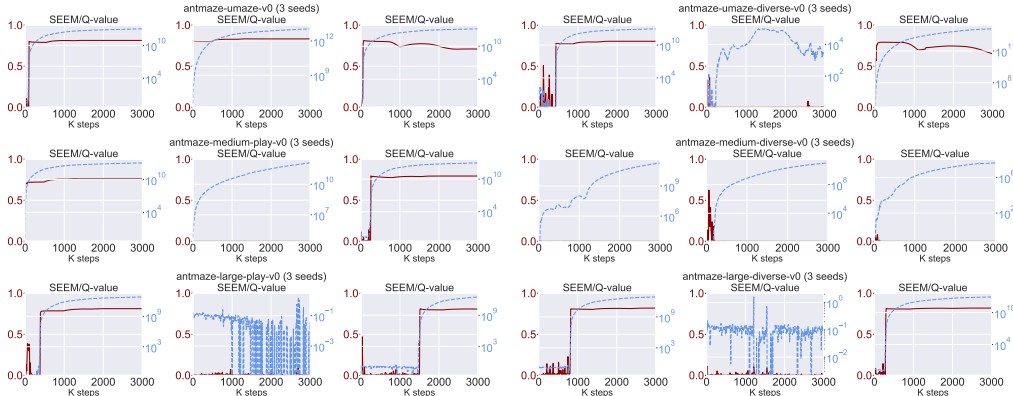

(a) The normalized kernel matrix's SEEM (in red) and the estimated Q-value (in blue) in D4RL Mujoco tasks.

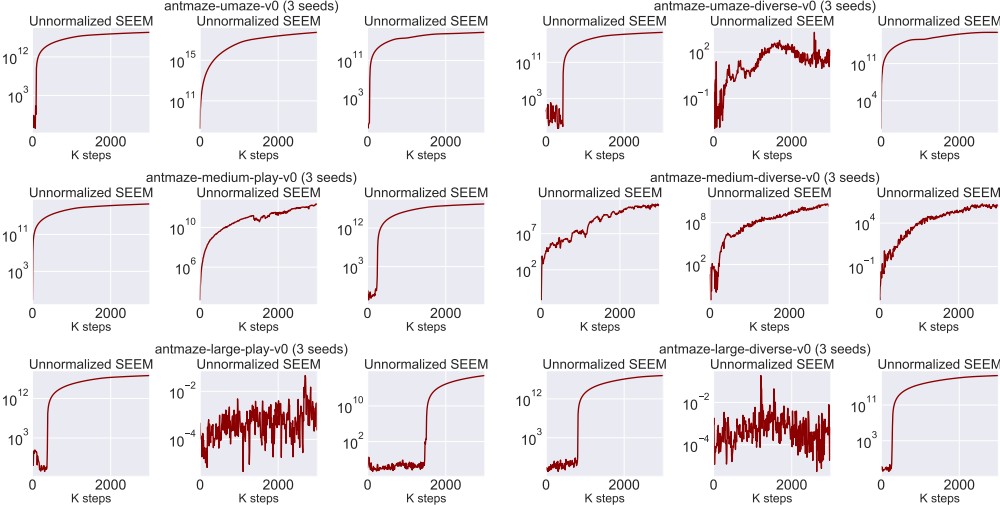

(b) The unnormalized kernel matrix's SEEM. The three curves in each environment correspond directly to those presented in Figure (a)

Figure 20: In Figure (a), an inflation in the estimated Q-value coincides with a surge in the normalized SEEM. However, there are some anomalies, such as the second running in the 'umaze-diverse' environment, where the Q-value rises while the unnormalized SEEM remains low. However, the corresponding normalized SEEM in Figure (b) suggests an actual inflation of SEEM. Furthermore, for scenarios where the Q-value converges, as seen in the second running in 'large-diverse', the unnormalized SEEM maintains an approximate zero value.

