# OpenReview forum: "Understanding, Predicting and Better Resolving Q-Value Divergence in Offline-RL"
_NeurIPS.cc/2023/Conference — NeurIPS 2023 poster_

### Official Review · Reviewer_4Xya · 2023-06-07

**Soundness:** 4 excellent
**Presentation:** 4 excellent
**Contribution:** 4 excellent
**Rating:** 7
**Confidence:** 4

**Summary:**

This paper studies the divergence phenomenon in Q-value iteration methods (e.g. Q-learning), especially focusing on the offline RL scenario. They introduce a theoretical framework for studying this issue, predicting divergence and even the training step at which it is likely to happen. Such analysis, which is based on the NTK (Neural Tangent Kernel), fully contains the linear case study (i.e. deadly triad analysis in linear settings) and extends it to non-linear value approximation. The insights they draw allow them to show that reducing a new metric, SEEM which measures self-excitation using NTK, reduces divergence by regularising the network. Regularising the generalisation of Q-networks as opposed to the dominant view of constraining the policy results in a new viewpoint in handling divergence in Q estimation and could yield many improvements in this domain in the future. They experiment with various regularisation methods from the deep learning toolbox, including dropout, BatchNorm, and LayerNorm, showing that LayerNorm is best able to stabilise training (i.e. avoid blowup in Q estimates) and achieve more homogenous and lower SEEM. They show that BC (behaviour cloning) also achieves a low SEEM but at the cost of bias due to policy constraints, which does not allow it to find the best solution or perform well in more challenging domains.

**Strengths:**

- The regularisation viewpoint on this phenomenon is a new one (as far as I know) yet a natural one.

- The regularisation viewpoint frees us from introducing policy constraints (which introduce bias) and allows us to simply solve the problem using off-the-shelf methods from the deep-learning toolbox.

- The theorems (if proofs are correct, I couldn't verify them personally with certainty) bring highly beneficial theoretical insights to the problem.

- The experiments are remarkable at showing that: (1) SEEM is a powerful metric in signifying divergence, (2) LayerNorm reduces SEEM and more homogeneously does so during training than other regularisation methods, resulting in stable Q estimation, (3) LayerNorm combined with SOTA offline RL methods results in significant improvements, especially under scarce access to offline data.

**Weaknesses:**

- I believe a short description of offline RL methods used could enhance the exposition of the ideas: policy constraint is not formally introduced because the methods incorporating it are not discussed in any depth.

**Questions:**

1. Given that this paper does not consider EMA or frozen targets (as in DQN), I am wondering how much using such gradual target updates contributes to alleviating divergence in Q estimation. In relation to the above question, I'm also uncertain how much using double Q-learning alleviates such issues in the offline RL setting. Wouldn't it be useful to have an ablation study to see, e.g., how much adding each of these in combination with each other helps performance and how much each of these help in isolation? Say, use **DDPG w/o EMA** vs. **DDPG w EMA** vs. **DDPG w EMA and Double Q-learning** vs. **DDPG w EMA and Double Q-learning and LayerNorm**, showing that additional advancement improves performance. Or compare in isolation, e.g.: **DDPG w/o EMA w LayerNorm** vs. **DDPG w EMA w/o LayerNorm**.

2. Are the results in Fig. 9 in combination with EMA? If not, what if you use EMA to show how the insights from the scenario w/o EMA carry over to the with-EMA case?

3. Does using LayerNorm as in your solution have the potential of allowing online DQN-style methods to step away from EMA/frozen target approaches altogether towards using the same online and target networks?

4. Does your solution allow DQN to solve Baird’s Counterexample (which was first used to show the potential divergence of Q-learning with function approximation)?

5. Does the analyses apply also to discrete-action pure critic settings such as DQN?


**Minor:**

- Can you elaborate on the way BC introduce explicit bias?

- I assume from the size of the $d_0$ layer that the architecture of the Q-function is action-in, correct?

Line 82: don’t -> do not

Line 119: To summary -> In summary

Line 123: explains -> explain

**Limitations:**

1. Analysis does not directly apply to the standard practice of using Exponential Moving Average (EMA; as in DDPG) or target-network freezing (as in DQN).

2. It is not clear how much of the contributions would carry over to the online learning setting and what would be the implications of using LayerNorm in online RL.

3. The analysis seems to apply to MLP architectures. It is not directly discussed how much of the contributions would apply to commonly used architectures such as ConvNets, ResNets, etc.

---

> ### Author Rebuttal · Authors · 2023-08-08
>
> Thank you for your time and constructive feedback. Please see the response below.
> # Q1: Clarification on EMA, Double-Q, and LayerNorm's Contributions
> Your suggestion to illustrate the interplay of EMA, Double-Q, and our LayerNorm in mitigating Q-value divergence is particularly insightful. To this end, we conducted an ablation study in the following sequence: "DDPG without EMA", "DDPG + EMA", "DDPG + EMA + DoubleQ", and "DDPG + EMA + DoubleQ + LayerNorm". Each successive step built upon the last. We carried out these experiments in settings of 10% and 50% Mujoco - environments notably susceptible to divergence. We report the total scores across nine tasks; please refer to Figure 1(a) in the attached PDF in global response for specific results.
> It clearly reveals that the EMA and double-Q, which are used as common pracitce in popular offline RL algorithms such as CQL, TD3+BC contribute to preventing diverngence and thus boosting performance. Nevertheless, **built upon EMA and DoubleQ, our LayerNorm can futher significantly boost the performance by surpressing divergence.**
>
> # Q2: Whether results in Fig. 9 use  EMA?
> You're correct. Fig.9, as well as all experiments in Section 5 (Performance Evaluation), **incorporate EMA**. This is in line with previous practical algorithms such as TD3, CQL, and IQL. We conducted a theoretical analysis without EMA to reduce the complexity of the derivation. Nevertheless, the significant outcomes presented in Section 5 imply that insights derived from scenarios without EMA still apply effectively when EMA is included.
>
> # Q3: LayerNorm Allows Online Methods without EMA
> Upon your suggestion, we tested SAC without EMA in two environments in **online settings**: Hopper and Walker. Please refer to Figure 2 in the attached pdf for the curves of return. Surprisingly, we discovered that **LayerNorm solution allows the SAC without EMA to perform equivalently well as the SAC with EMA**. In contrast, SAC without EMA and LayerNorm behaved aimlessly, maintaining near-zero scores. It reveals LayerNorm (or further regularizations discovered by SEEM later) have the potential of allowing online DQN-style methods to step away from EMA/frozen target approaches altogether towards using the same online and target networks.
>
> Your advice led us to these unexpected yet intriguing findings. These results partially reveals potential implications of SEEM in an online learning setting, as written in the limitations section of your review. We leave SEEM in online setting as future work.
>
>
> # Q4: Baird’s Counterexample
> We would like to clarify that Baird's Counterexample specifically targets **linear** approximation. In our experiment, we found **neural networks** exhibited stable convergence within the Baird counterexample scenario. Furthermore, we investigated cases of divergence within linear situations and discovered that **incorporating LayerNorm in front of linear approximation could effectively lead to convergence** (see Fig. 1(b) in the attached pdf). This finding is not entirely surprising. Since a linear regression model can be considered a special case of neural networks, our analysis can be directly applied to linear settings.
>
> # Q5: Discrete-action pure Critic Settings
> Indeed, our analyses are not limited to continuous scenarios; they are equally applicable to discrete settings. The theoretical insights can be seamlessly transferred to a discrete setting, requiring only negligible alterations. **We have verified our analysis using a simple discrete-action gridworld with DQN**. Notably, in the gridworld scenario, all conclusions mentioned in the paper are observed, including the Q-value divergence in sync with the rise of SEEM, linear decay of Q-value's inverse with the SGD optimizer, and etc. We've also assessed LayerNorm's effectiveness in curbing divergence in the discrete gridworld. Please do not hesitate to inquire further on this topic.
>
>
> # Response to Minors
> [The way BC introduces explicit bias] **Simply speaking, BC restricts any learned policy to be near the behavior policy. This bias is harmful when the behavior policy is sub-optimal**. An illustration is the example in Fig. 7 In our paper. Specifically, the offline dataset of Antmaze-large-play features behaviors of various qualities. When employing a strong BC (with a large BC coefficient of 3 or 10) to circumvent OOD actions and subsequent divergence, the learned policy is forced to mimic some sub-optimal actions, resulting in poor performance.
>
> [Architecture of the Q-function] You are correct. The Q-function is action-in.
>
> # Conclusion
> In conclusion, we appreciate the opportunity to address these vital concerns. We hope that our responses sufficiently clarify your question.  We would be grateful for your reconsideration of the **confidence**, in light of the explanations provided. If there are any additional questions about our research, please do not hesitate to reach out.

---

> > ### Comment · Reviewer_4Xya · 2023-08-19
> >
> > Thank you for your response, thoroughly addressing my questions/concerns. I also read through the discussion with reviewer T4o6 and believe that the reviewer's concerns are being addressed. I have now raised my confidence score and would be glad to see the paper accepted.

---

> > > ### Author Response · Authors · 2023-08-19
> > >
> > > Thanks for taking the time to review our paper and for your prompt response. We also appreciate your constructive suggestions, which help us more clearly identify LayerNorm's effectiveness. We are honored to have received your support for its acceptance.

---

### Official Review · Reviewer_T4o6 · 2023-06-14

**Soundness:** 2 fair
**Presentation:** 3 good
**Contribution:** 2 fair
**Rating:** 5
**Confidence:** 4

**Summary:**

The paper theoretically investigates the problem of value function overestimation in offline RL through the lens of neural tangent kernel (NTK). Additionally, the paper presents empirical findings that validate the effectiveness of incorporating LayerNorm before each activation function in mitigating value network divergence. The paper conducts extensive experiments on D4RL AntMaze and D4RL MuJoCo Gym while varying the dataset size and demonstrates that LayerNorm ensures stable value convergence and leads to state-of-the-art performance even with significantly small datasets.

**Strengths:**

- The paper is well written and organized. The paper provides empirical evidences to support its theoretical claims (Figures 3, 4).
- The proposed method can be easily integrates with existing offline RL algorithms (e.g., CQL, IQL, TD3+BC) and consistently improves performance.

**Weaknesses:**

- It is worth noting that DR3 has already investigated the dynamics of Q-learning in offline RL using NTK [1]. It is crucial for the author to properly reference this prior work and to establish a clear connection between the two studies. The author should explicitly highlight the similarities and differences between the proposed analysis and the findings presented in DR3.
- The idea of applying LayerNorm (or GroupNorm) has already been proposed in Scaled QL [2].
- There should be a proper reference to support the claim that the neural network becomes a linear function when the input’s norm is too large (line 234).
- It is unclear why the linearity of the neural network leads to an improperly large kernel value (line 234).
- The experimental setup in Section 5.2 is very similar to DR3 [1].

[1] Aviral Kumar et al., DR3: Value-Based Deep Reinforcement Learning Requires Explicit Regularization, ICLR 2022. \
[2] Aviral Kumar et al., Offline Q-Learning on Diverse Multi-Task Data Both Scales And Generalizes, ICLR 2023.

**Questions:**

Please see the weaknesses above.

---

> ### Author Rebuttal · Authors · 2023-08-06
>
> Thank you for your time and thoughtful review. Please see the response below.
> # Linear NTK Value
> Please refer to the global response for a detailed explanation.
> # About Missing References
> Thank you very much for pointing out these two papers that we missed. Since we conducted literature research along the line of works about Q-value divergence and deadly triad in offline RL, we were unaware of these two papers when writing the draft. We apologize for such unintended oversight.
> After carefully re-examing these two papers, we agree that these two papers are very relevant to our work, hence we will add them to our related work section with discussions in all future versions.
>
> **However, despite similar observations, we would like to highlight some important differences between our work and these two references.**
> # Comparison with DR3
> Indeed, our work shares similar observations of the connection between Q-value divergence and feature rank collapse (called feature co-adaptation in DR3). However, our work is different in the following aspects.
> ## 1. More General Setting & Different Perspectives
> While DR3 attributes feature rank collapse to implicit regularization where $L(\theta)=0$, we propose a different perspective.
> To start with, we first find that
> * (i) Normalization-free networks' value prediction of the dataset sample and extreme point (i.e. action $A_{ex}$ with maximum value entries) exhibit a strange but strong correlation (having large NTK value)
> * (ii) Such networks also tend to output large values for these $A_{ex}$ (see global response), which makes these extreme points easily become policy actions. In Fig. 3 in global response pdf, you can see policy actions tend to move toward extreme points when Q-value divergence occurs in all D4RL tasks.
>
> It is then shown that (i) and (ii) are the root cause for many consequences, including feature co-adaptation and Q-value divergence.
> 1. The Q-value estimation network overestimates some out-of-sample (typically extreme point) action $A_{ex}$'s Q-value $\hat{Q}(s', A_{ex})$.
> 2. This inflated value becomes bootstrapped target, passing its influence to state $s$ by TD update, where the offline dataset contains transition tuple $(s,a,s')$. The Q-network is updated to boost $\hat{Q}(s,a)$.
> 3. Due to the large NTK value between $(s', A_{ex})$ and $(s,a)$ (see Fig. 5), TD updates the network's parameter to increase $\hat{Q}(s', A_{ex})$ even more than $\hat{Q}(s,a)$.
> 4. A loop is formed by 2 and 3. $\hat{Q}(s,a)$ keeps chasing $\hat{Q}(s', A_{ex})$'s value, leading the model's parameter to diverge along certain direction to infinity, which causes every input's feature becomes parallel.
>
> This framework is **not restricted to over-parametrized regime which requires $L(\theta)=0$, nor does it require the label noise $\varepsilon$ from SGD**. Actually, such near-zero critic loss assumption is mostly not true in the real practice of RL. Also, our analysis does not rely on any n-dimensional basis or sub-space assumptions.
>
> **Therefore, our analysis provides an alternative explanation for feature rank collapse and Q-value divergence from the perspective of norm-free network's pathological extrapolation. It applies to more general settings and provides new information.** These novel insights and implications are not covered in previous work, because it requires several non-trivial observations together with a rigorous theory to encapsulate them. As reviewer 4Xya said, our work "results in a new viewpoint in handling divergence in Q estimation and could yield many improvements in this domain in the future."
> ## 2. Divergence Metric & Precise Dynamics
> We established a measure (SEEM) that can synchronously signify divergence. **SEEM is not just an index to indicate divergence, but also a new handle to solve the problem**. Moreover, DR3 does not accurately characterize model's dynamics as our work does. We formally prove in Theorem 1, 2, 3 that the model's parameter $\theta$ **evolves linearly for the Adam optimizer** (see Fig 4 and 14), and **collapses at a certain iter for SGD** by solving an ODE (see Fig 3). This accurate prediction demonstrates our precise understanding of neural network dynamics and is absent in previous work like DR3 and Scaled-QL.
> ## 3. Simpler and Cheaper
> DR3 requires hyperparameter tuning for $c_0$, which involves more effort in searching proper values across different environments. Moreover, it introduces extra computation when computing the gradient of $\phi(s',a')$ to get $\overline{\mathcal{R}}_{exp}(\theta)$. Our method is free of all these shortcomings by simple layernorm.
> # Comparison with Scaled-QL
> Thank you for mentioning this work. Indeed, as we acknowledged in Line 242-247, some previous works like LB-SAC, RLPD have empirically utilized LayerNorm. We will further add Scaled-QL to reference. Nevertheless, compared to Scaled-QL, our contribution is three-fold:
> ## 1. Theoretical Justification
> We provide rigorous theoretical justification for why normalization helps where previous works do not. **The fundamental mechanism of Q-value divergence and its relation to the extrapolation of Q-networks are studied.**
> ## 2. Comprehensive Empirical Evaluation
> The sole application of Scaled QL on CQL doesn't show LayerNorm's effectiveness for other popular offline RL algorithms, while our research confirms the efficacy of regularization methods that can diminish SEEM and improve performance **across a range of offline RL algorithms like CQL, IQL, TD3+BC, and Diffusion-QL, including the primary classes of offline RL.**
> ## 3. Beyond Normalization
> As the other reviewers acknowledged, the SEEM metric "will lead others to further engineering that will yield even better-performing adjustments than the simple addition of the LayerNorm". With the SEEM metrics, **we can examine regularization methods from the DL toolbox and even create new methods.** This could potentially inspire further advancements in the field.

---

> > ### Comment · Reviewer_T4o6 · 2023-08-11
> >
> > Thank you for your explanation on the differences between the proposed analysis and that of DR3. While Theorem 3 in your paper and Theorem 3.1 in DR3 arrive at similar conclusions, DR3 adopts more assumptions. I find this quite intriguing. Have you verified the validity of the assumptions and lemmas you made? Specifically, regarding Lemma 1, it is not common that an MLP with ReLU activation is a homogeneous function in the parameter space.

---

> > > ### Author Response · Authors · 2023-08-11
> > >
> > > Thank you for your time and reply. Here is our further explanation, we hope this can help to resolve your confusion.
> > > # About Theorem 3 and Theorem 3.1 in DR3
> > >
> > > Yes, these two results do seem to be similar. But they are actually derived from quite different foundations. Our results convey novel implications by capturing a mechanism that is more realistic, so that it requires fewer assumptions to reach a similar conclusion.
> > >
> > > Specifically, DR3 Theorem 3.1 analyzes the condition for $\theta^*$ to be the stable fixed point of TD-update, namely $Q_{\theta^*}\left(s_i, a_i\right)=r_i+\gamma Q_{\theta^*}\left(s_i', a_i' \right)$ for every $(s_i,a_i,s_i')$. This means DR3's analyzing framework **focuses on the setting where the model perfectly fits the offline training data**, and the SGD noise plays a role as implicit regularization. This noise continues to increase the Q-value for the unseen data point $Q_{\theta^*}(s_i',a_i')$ in the null space of $G$, while preserving its fixed-point property in training data at the same time. As a consequence, all the dataset points' value inflates to infinity.
> > >
> > > Our key difference and advancement is that, we found such divergence does not happen in the parameter subspace where the Q-network perfectly fits the training data's Q-value Bellman equation. Actually, the model is "chasing its shadow" rather than staying at the stable fixed point of Bellman update. Once the pathological extrapolation of MLP starts the so-called self-excitation procedure, **the training loss will never become near-zero, but keep exploding**. The underlying mechanism is explained in our initial rebuttal response from step 1 to 4. Each TD update step on $\theta$ tries to let $Q_{\theta}(s_i, a_i)$ get closer to $r_i+\gamma Q_{\theta}\left(s_i', a_i'\right)$, but increase the latter bootstrpped target even more due to improper generalization.
> > >
> > > In conclusion, our theory for explaining the divergence mechanism differs at the very beginning. It captures a mechanism that has yet to be fully discovered before. Therefore, being more fundamental, our analysis is able to reach a conclusion without assumptions like zero training loss or assumptions on SGD noise's covariance, etc.
> > >
> > > # Homogeneity of ReLU-activate MLP
> > >
> > > Indeed, ReLU-activated MLP **with bias** is not rigorously homogeneous in the entire parameter space. But bias-free ReLU activated MLP, namely function $f(x)=W_L \sigma(W_{L-1} \sigma (\cdots \sigma(W_1 x)))$ is homogeneous with respect to $\theta=(W_1, W_2, \cdots, W_L)$. You can verify this from simple fact that $W_i x$ and ReLU activation $\sigma(\cdot)$ are homogeneous functions ($\max(kx,0)=k\max(x,0)$), and the composition of homogeneous functions is homogeneous.
> > >
> > > However, **it is important to point out that our analysis also applies to ReLU-activated MLP with bias in each layer.** The main reason is a little subtle. We empirically found that such homogeneity still holds with high precision for MLP with bias, at least good enough in signifying the value divergence and giving correct asymptotic order. We hypothesize that when value divergence is going to happen, the product dot $W_i z_i$ is relatively large compared to the scalar $b_i$,  and the effect of bias in output is negligible.
> > >
> > > We also run experiments to validate the homogeneity. We define a family of 3-layer ReLU-activated MLPs (with bias term) with different scaling $\lambda$ of the same network parameter $\theta$ (from D4RL training checkpoints when value divergence happens), and feed these networks with the same input. We show one example in the below table, which confirms that the $L$-degree homogeneity of output, gradient and NTK is valid at high precision, the NTK is almost parallel, too. The larger the scaling factor is, the more accurate $k^L$ increasing pattern is for output ( $k^{L-1}$ for gradient and $k^{2(L-1)}$ for NTK), as described in Lemma 1. Further, we empirically validated the $L$-degree homogeneity of NTK holds for all checkpoints in D4RL experiments where divergence happens.
> > >
> > > |  $\lambda$   | 5 | 10 |
> > > |  ----  | ----  | ---- |
> > > |  $f_{\lambda \theta}(x)/f_{\theta}(x)$ | 124.99 ($5^3$) | 999.78 ($10^3$) |
> > > |  grad scale  |  25.001 ($5^2$) | 100.001 ($10^2$) |
> > > |  NTK scale  |  624.99($5^4$) | 9999.84 ($10^4$) |
> > > | NTK cos | $>1-10^{-6}$ |  $>1-10^{-6}$ |

---

> > > > ### Comment · Reviewer_T4o6 · 2023-08-12
> > > > **Please check the lemma**
> > > >
> > > > I appreciate your response. It is now evident that Lemma 1 relies on empirical observations rather than being theoretically accurate. Now, let's look at Theorem 1. Is it appropriate to approximate $X_{t+1}^* \approx X_t^* $, knowing that $X_t^*$ contains the $\arg\max$ function? It is important to carefully examine the proofs of the lemmas and theorems you proposed, considering their informal nature.

---

> > > > > ### Author Response · Authors · 2023-08-12
> > > > > **Clarifying and Validating Assumptions**
> > > > >
> > > > > Thank you for your appreciation. We highly value your comments and believe that they have allowed us to further refine our work.
> > > > >
> > > > > Firstly, we would like to clarify the expression regarding our paper's reliance on empirical observations rather than being  theoretically accurate. **Our approach aims to be theoretically accurate and rigorous under the realistic assumption**. This assumption that the bias term has a negligible effect on output and NTK in Q-value divergence scenarios is backed by empirical validation, as we have shown. **Adopting realistic assumptions does not undermine the value and accuracy of theoretical studies.** For example, DR3 also adopts a zero-critic loss assumption, which might not always hold true in real offline RL scenarios.
> > > > >
> > > > > Regarding the equation $X^*_{t+1} \approx X^*_t$, we agree that validating this assumption is crucial. Indeed, we were intrigued when we first discovered this relationship. **We have verified it in all offline environments we studied, and showcase one of them in our draft**. You can refer to Fig 1 in our draft, where the second column "action similarity" measures the cosine similarity between each step's policy action and the final policy, remaining 1 for most of the time. Our attachment to the global response also shows that policy actions become extreme-point actions once Q-values diverge. These two experimental results support our assumptions.
> > > > >
> > > > > Your concerns were helpful for identifying unclear justifications for assumptions and  improving the clarity of our paper. We will revise our paper accordingly to make the validation of assumptions more clear. We're grateful for your reviews.

---

> > > > > ### Author Response · Authors · 2023-08-17
> > > > > **Enhancing Precision of Assumption Statement**
> > > > >
> > > > > After carefully considering your concerns regarding our assumptions, such as $X_{t+1}^* \approx X_{t}^*$ and the homogeneity of the ReLU-activated MLP, we realized the potential ambiguity introduced by our informal notation "$\approx$". This could potentially compromise the rigor of our theoretical deductions. While our assumptions have been empirically validated in pratical experiments, we understand your concern about the rigor of the assumption and theoretical preciseness. We agree that **using precise quantification such as** $|X_{t}^* - X_{t+1}^*|=o(\eta)$ **rather than** $X_{t+1}^* \approx X_{t}^*$ **would provide a clearer description of our assumptions**. Similarly for the definition of near-homogeneity of ReLU MLP. Adopting such formal notation will refine our proofs and make our study more precise. We will revise our theory statement in a more rigorous manner.
> > > > >
> > > > > We really appreciate your constructive feedback about the rigorous assumption definition. Indeed, a formal description is essential for enhancing a theorem's accuracy. However, we believe this lack of formalism doesn't invalidate the correctness of our theorem or diminish the overall contribution of our work.  The identified issues related to these assumptions are technical in nature, rather than fundamental flaws.  Since the assumptions are aligned with empirical observations and the subsequent deductions are carried out with rigor, our results rest on a solid, realistic foundation. We have verified that all theoretical results can also hold with a rigorous assumption description, (e.g. o(·), O(·)) without essential difficulty.  We believe that the current limitations in the formal rigor of our assumptions will not detract from our vital contributions, specifically the novel insights into Q-value divergence and its underlying mechanism.
> > > > >
> > > > > Once again, thank you for your invaluable insights which have guided us to refine our theoretical presentation. We welcome any further discussions or clarifications.

---

> > > > > > ### Comment · Reviewer_T4o6 · 2023-08-19
> > > > > > **Thank you**
> > > > > >
> > > > > > Apologies for the delayed response. Many of my concerns have been addressed. However, it is not accurate to claim that $|X_{t+1}^* - X_{t}^*| = o(\eta)$ without any underlying assumptions, especially given that $\arg\max$ is not Lipschitz continuous. Upon reviewing both the main text and the supplementary materials, I was unable to locate any assumptions pertaining to Theorem 1. In the revised version, please explicitly specify the assumptions you've employed to validate the theorems. I raise my score from 3 to 5.

---

> > > > > > > ### Author Response · Authors · 2023-08-19
> > > > > > > **Thanks for your response**
> > > > > > >
> > > > > > > Thank you for for taking the time to thoroughly review both our main text and supplementary materials. We greatly appreciate your expert suggestions, which indeed have helped us improve the rigor and clearness of our paper. We will ensure to explicitly specify these assumptions pertaining to theorems in the revised version of the manuscript.  We're also grateful for your increased score and your overall appreciation of our paper.

---

### Official Review · Reviewer_WzD5 · 2023-07-05

**Soundness:** 4 excellent
**Presentation:** 3 good
**Contribution:** 4 excellent
**Rating:** 7
**Confidence:** 4

**Summary:**

This paper analyzes Q-value divergence in Offline-RL by considering a
neural tangent kernel for the value function.  They show that
consideration of this kernel is predictive of Q-value divergence.
This analysis further leads to the observation that using a LayerNorm
yields a kernel that behaves more like one would hope -- with nearer
values being more impacted than farther ones.  Their empirical tests
on D4RL benchmarks similarly show the benefits of LayerNorm.


**Strengths:**

I like the insights in this paper and suspect that its publication
will lead others to further engineering that will yield even better
performing adjustments than the simple addition of the LayerNorm.
Figure 5 suggests there is more to be done.  Why does the figure on
the left indicate almost the opposite of what one would hope for.  At
a minimum, should the value right at x_0 be red?


**Weaknesses:**

The paper needs a number of editing improvements.  First, it needs an
English grammar checker.  The errors mostly don't lead to difficulties
in understanding, but, the paper should not be published with the
current level of English quality.

In section 5.1, the BC term should be explained even though the
reference is given for it.

The reference to a score of 81.4 in that section does not appear to match
the numbers in Table 1.

The legends in the figures need to use a larger font size.



**Questions:**

None

---

> ### Author Rebuttal · Authors · 2023-08-08
>
> Thank you for your time and thoughtful review. On your questions, please see the response below.
> # Explanation for Figure 5
> Please note that the color in Figure 5 represents the normalized NTK. Indeed, in the left figure, the absolute NTK surrounding $x_0$ is positive. However, values farther from $x_0$ are significantly larger. As a result, the NTK around $x_0$ is minimal and normalized to a value close to zero, as indicated by the blue color.
> # Addressing  Weaknesses
> We appreciate your suggestions on grammar, font size, and other issues and will correct them in our formal version. We will conduct a thorough grammar check to correct grammar and spelling mistakes. Additionally, we intend to provide a formal introduction to BC and adjust the font size of the legends in subsequent versions of the paper.
>
> The value 81.4 in Line 289 is a previous result obtained using fewer seeds, whereas 80.8 in the table represents the accurate result computed with 10 seeds. We will rectify this discrepancy.

---

### Author Rebuttal · Authors · 2023-08-08

**Note**: We appreciate the constructive feedback from all reviewers. We have prepared a PDF document for reviewers, containing figures about extensive experiments and results that further illustrate and support our response in the rebuttal.

# Why linear function and large NTK value?
Some reviewer raises questions about why ReLU-activated MLP without norm layer becomes a linear function when the input's norm is too large, and why the linearity of the neural network leads to an improperly large kernel value. Here is a more detailed explanation with intuition.
- **Linearity Prediction Outside Dataset Range**

As has been pointed out in [1] (Fig. 1), ReLU-activated MLP without any norm layer becomes a linear function for the points that are excessively out of the dataset range. This fact can be comprehended in the following way: Consider an input $\lambda x$, the activation state for every neuron in the network becomes deterministic when $\lambda \to \infty$. Therefore, the ReLU activation layer becomes a multiplication of a constant 0-1 mask, and the whole network degenerates to a linear function.
- **Large NTK Value between extreme point and dataset sample**

If the network becomes a linear function for inputs $\lambda x_0, \lambda\geq C$, this will cause an arbitrarily large NTK value between extreme points $\lambda x_0$ and dataset point $x_0$. Because now $f_{\theta}(\lambda x)$ can be written in equivalent form $W^T (\lambda x)$, so $\phi(\lambda x_0)=\nabla_{W}W^T (\lambda x_0)= \lambda x_0$ becomes linear proportional to $\lambda$. This further induces linear NTK value between $x$ and $\lambda x$, which can be unbounded.

This phenomenon can be intuitively interpreted as below, since the value prediction of $f_{\theta}(x)$ on line $\lambda x_0$ is almost a linear function $W^T x$ for large $\lambda$, any subtle change of effective parameter $W$ will induce a large change on $\lambda x_0$ for large $\lambda$, proportional to $||x||_2$. This demonstrates a strong correlation between the value prediction between point $x_0$ and far-away extreme point $\lambda x_0$. The rigorous proof can be found in Proposition 1.

Note that such strong correlation is not restricted to rigorous paralleled inputs $x_0$ and $\lambda x_0$, but also between many dataset small-norm input $x_0$ and extreme large-norm input $x_{ex}$, once they share similarly activated neurons and effective linear parameters. The so-called "extreme" points' norm does not need to be several magnitudes larger than dataset point $x_0$ either. It just needs to be constantly larger to be able to start the self-excitation procedure, therefore starting the divergence loop.


[1] How Neural Networks Extrapolate: From Feedforward To Graph Neural Networks. Xu et al.

---

### Decision · Program_Chairs · 2023-09-21

**Decision:**

Accept (poster)

**Comment:**

This paper brings an insightful analysis on the Q-value estimation divergence in offline RL, proposes a new metric based on NTK, SEEM, to measure the divergence, and suggests using regularization to mitigate the problem, among which LayerNorm shows best performance in the experiments.

All reviewers appreciate the insights from their theoretical analysis. Empirical evaluation shows the efficacy of the new metric and justifies the benefit of LayerNorm in Q-value estimation.

The main concerns come from the following aspects:
- Review T4o6 points out several flaws in the proof of theory and lemma. After discussion, all agree that the problem can be fixed by clarifying additional assumptions and improving the rigorousness of the proof, which authors have promised to add in the revision.
- Similarity to the work of DR3 (Aviral Kumar et al., ICLR 2022) both of which conduct analysis on the dynamics of Q-value estimation in offline RL and suggest explicit regularization as a solution. The authors first admit the ignorance of that reference in the initial submission and then explain the different perspective of the analysis in the two works and different assumptions applied. Review T4o6 agree with the comparison though remain suspicious if the assumptions used in DR3 is more restrictive than this work. It would be very important for the authors to provide detailed comparison in the revision and, if possible, include DR3 as an additional baseline in the experiments.
- LayerNorm has been used in previous works such as Scaled-QL (Aviral Kumar et al., ICLR 2023) and those already mentioned in the submission. The authors explain that this submission provides theoretical justification to the use of LayerNorm besides other regularizations in contrast to empirical usecases in other works and they provide more comprehensive empirical evaluation.

After the discussion, all reviewers agree on acceptance of this submission. The remaining concerns should not impact the theoretical contribution of this work after the additional promised edits. I strongly encourage the authors to take into consideration all the discussion in the reviewing process, fix the proofs and make a final revision according accordingly.